# Multi-scale Minimal Sufficient Representation Learning for Domain Generalization in Sleep Staging

## Abstract

Deep learning-based automatic sleep staging demonstrates strong performance as a promising solution for diagnosing sleep disorders. However, deep learning models often struggle to generalize on unseen subjects due to variability in physiological signals, resulting in degraded performance in out-of-distribution scenarios. To address this issue, domain generalization approaches have recently been studied actively to ensure generalized performance on unseen domains during the training. Among those techniques, contrastive learning has proven its validity in learning domain-invariant features by aligning samples of the same class across different domains. Despite its potential, many existing methods are insufficient for extracting truly domain-invariant representations, as they do not explicitly reduce domain-relevant information embedded in the features. In this paper, we argue that addressing superfluous information is a key to bridging the domain gap. Furthermore, existing methods often neglect the multi-scale nature of sleep signals, potentially missing important temporal and spectral characteristics. To address these limitations, we propose a novel Multi-Scale Minimal Sufficient representation learning (MSMS) framework, which effectively reduces domain-relevant information while preserving essential temporal and spectral features for sleep stage classification. We evaluate our method on publicly available sleep staging benchmark datasets, SleepEDF-20 and MASS. Experimental results demonstrate that our approach consistently outperforms state-of-the-art methods.

## 1 Introduction

Sleep staging, the process of identifying and tracking transitions between different sleep stages over time, plays a pivotal role in analyzing sleep quality and treating sleep disorders (Scott et al., 2023). Typically, experts categorize sleep states into five stages—Wake, N1, N2, N3, and rapid eye movement (REM)— using polysomnography (PSG), which records various physiological signals. While manual sleep staging remains the gold standard, it is both labor-intensive and time-consuming, often requiring trained specialists to carefully examine hours of physiological data. To alleviate these challenges, deep learning (DL)-based techniques have emerged as a powerful alternative. Despite such advanced, numerous DL-based techniques inevitably struggle when confronted with out-of-distribution (OOD) data (i.e., unseen domain), leading to significant performance degradation caused by a discrepancy in data distribution (Zhou et al., 2022).

The challenge of OOD generalization in sleep staging is particularly prevalent due to the high variability in physiological signals between individuals. For instance, insomnia patients typically exhibit increased high-frequency activity and reduced slow-wave sleep in electroencephalogram (EEG) signals, which measure brain activity (Buysse et al., 2008). Moreover, age-related changes add to this complexity; research has shown that slow-wave sleep decreases with age—by as much as $2\%$ per decade in adults—while the proportions of N2 and REM sleep undergo significant shifts across the lifespan (Ohayon et al., 2004). These patient-specific characteristics, or covariates, pose a significant challenge for DL models, often causing them to perform poorly on data from unseen subjects.

Domain generalization (DG) aims to enhance the robustness of DL models by improving their ability to generalize across unseen data domains. Prior works in DG have focused on learning domain-

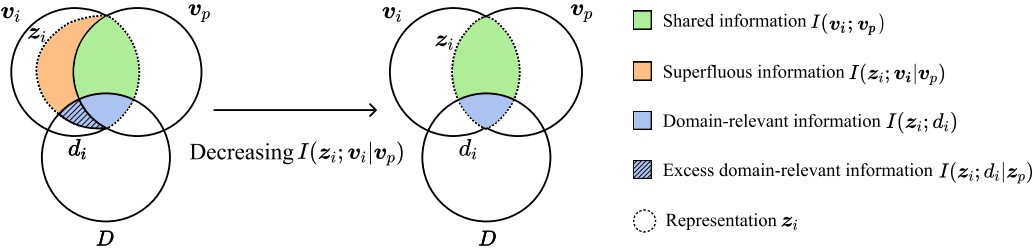

(a) Sufficient representation    (b) Minimal sufficient representation

Figure 1: Comparison between sufficient representation and minimal sufficient representation. In conventional contrastive learning, $z_i$ denotes the normalized feature of $i$-th sample in the batch, while $z_p$ represents the normalized feature of positive sample $v_p$ for $v_i$ (a sample with the same label as the $i$-th sample). (a) Sufficient representation: In this paradigm, the feature representation $z_i$ (illustrated by ellipse with dashed line) includes superfluous information $I(z_i; v_i|v_p)$, which is biased towards the specific characteristics of the $i$-th sample. It is not effective in learning domain-invariant features because excess domain-relevant information $I(z_i; d_i|z_p)$ remains within the features. Here, $d_i$ refers to specific domain information in domain factor $D$ that is associated with $v_i$. (b) Minimal Sufficient Representation: In contrast, minimal sufficient representation learning aims to reduce the superfluous information $I(z_i; v_i|v_p)$, thereby diminishing the domain-relevant information within the feature representation (illustrated in light blue in the figure). This reduction enables the model to learn domain-invariant features effectively.

invariant features by aligning multiple source domains (Li et al., 2018c; Mahajan et al., 2021; Lu et al., 2022; Dayal et al., 2024). Within this paradigm, contrastive learning-based DG techniques have recently emerged as a promising strategy for extracting domain-invariant representation (Mahajan et al., 2021; Yao et al., 2022; Liu et al., 2023). These methods effectively align multiple domains by clustering samples of the same category (i.e., class) from different domains while simultaneously pushing apart dissimilar ones (i.e., negative pairs). Notably, those methods have demonstrated effectively learning generalized representations from biosignals, suggesting their potential applicability in sleep staging (Zhang et al., 2022; Wang et al., 2024b).

Contrary to their superiority, those approaches often struggle to extract genuinely domain-invariant representations. As illustrated in Figure 1(a), these methods primarily focus on increasing the shared information between positive samples, thereby facilitating sufficient representation learning, where the learned features retain all task-relevant information. However, this approach does not effectively eliminate domain-relevant information, which often remains embedded within the features as superfluous information-unshared information across different samples (Federici et al., 2020). For this reason, minimal sufficient representation learning, which aims to minimize superfluous information, is a crucial approach for achieving robust domain-invariant representations in DG. However, this approach risks overfitting the features of the final layer, potentially reducing the diversity of learned representations. This limitation is particularly significant in sleep staging tasks, where multi-level features are essential for capturing distinct frequency characteristics. More advanced methods are required to address these limitations, incorporating both the elimination of superfluous information and the utilization of multi-scale learning to preserve feature diversity and effectively capture hierarchical representations.

In this work, we propose a novel framework called Multi-Scale Minimal Sufficient representation learning (MSMS), designed to leverage multi-scale domain-invariant features to effectively bridge distribution gaps. The primary objective of our MSMS is to minimize domain discrepancies by reducing superfluous information via minimal sufficient representation learning. We argue that minimizing this superfluous information is crucial for extracting more robust domain-invariant features, as domain-relevant characteristics are still present in it, as illustrated in Figure 1(b). To mitigate potential information reduction and to enhance the model's capabilities for capturing diverse temporal and spectral characteristics inherent in sleep signals, we apply the proposed objective function across encoder features extracted from multiple layers. Consequently, the main contributions of our work are:

- To the best of our knowledge, we introduce a theoretically grounded novel objective function to learn minimal sufficient representations, providing a more effective method for domain generalization compared to traditional contrastive learning techniques.

- We proposed a novel integration of multi-scale learning within the minimal sufficient learning, effectively preventing overemphasis on specific layer features and enhancing generalization across domains.

- We demonstrate the superiority of our MSMS over state-of-the-art approaches on two sleep staging datasets, achieving significant improvements.

## 2 Related Work

### 2.1 Sleep Staging

Sleep staging refers to the classification of sleep states, which is crucial for assessing sleep quality and diagnosing sleep disorders (Melek et al., 2021). Many DL methods have been developed to classify sleep stages using PSG. Conventional DL approaches focused on CNN-based encoder architectures designed to effectively capture the temporal characteristics of EEG signals (Tsinalis et al., 2016; Supratak et al., 2017). Recent studies have introduced techniques that enable models to learn representations across multiple scales of the encoder, effectively reflecting diverse temporal and spectral characteristics from different perspectives (Eldele et al., 2021; Wang et al., 2022b; Lee et al., 2024). For example, Eldele et al. (2021) developed a multi-resolution CNN leveraging varying filter widths to capture features across multiple scales effectively. Similarly, Lee et al. (2024) proposed SleePyCo, which employed contrastive learning and a feature pyramid to capture multi-level features, which were then utilized in a transformer-based classifier. Despite these advancements, previous approaches often fail to generalize effectively to unseen subjects due to inadequately addressing variability in physiological signals across individuals. To overcome this limitation, our MSMS method proposed extracting subject agnostic features via the minimal sufficient representation learning.

### 2.2 Domain Generalization

Domain generalization techniques have been introduced to enhance model performance on unseen domains (Li et al., 2018a; Arjovsky et al., 2019; Xu et al., 2021). A common strategy in these approaches is to learn domain-invariant representations by aligning samples from different source domains (Volpi et al., 2018; Ding et al., 2022; Liu et al., 2024). For example, Li et al. (2018b) introduced a model that learns invariant features by considering the changes across conditional distributions over labels. Yao et al. (2022) utilized proxy-based contrastive learning to acquire domain-invariant representations by facilitating effective domain alignment. Dayal et al. (2024) introduced margin-based adversarial learning that uses margin loss-based discrepancy to learn domain-invariant features. Building on these advancements, several studies have investigated the application of domain generalization to sleep staging tasks, aiming to OOD challenges (Jia et al., 2021; Yang et al., 2023; Wang et al., 2024a). For instance, Yang et al. (2023) proposed a novel framework that uses mutual reconstruction and orthogonal projection techniques to extract domain-invariant features, addressing subject variability. Wang et al. (2024a) proposed a method for obtaining domain-invariant features through both epoch-level feature alignment and sequence-level alignment. Despite this superiority, they often overlook the importance of capturing both temporal and spectral information concurrently. Unlike the existing methods in the literature, our MSMS effectively captures both temporal and spectral information while ensuring domain-invariant representations by reducing superfluous information across multiple feature levels simultaneously.

## 3 Preliminaries

Contrastive learning aims to learn robust representations by enhancing the similarity between views of the same sample. In this context, views refer to different augmentations applied to the same input sample, which retain essential semantic information while enhancing input diversity. Let $v_1$, $v_2$, and $z_1$, $z_2$ represent two different views of the input sam-

ple $\boldsymbol{x}$ and normalized vectors of the projection head outputs from each view, respectively. Here, the projection head is typically a multi-layer perceptron to map low-dimension space. The relationships between $\boldsymbol{x}$, $\boldsymbol{v}_1$, $\boldsymbol{v}_2$, $\boldsymbol{z}_1$ and $\boldsymbol{z}_2$ are depicted in Figure 2, represented through a graphical model. The contrastive learning loss is designed to align the representations $\boldsymbol{z}_1$ and $\boldsymbol{z}_2$, ensuring they retain consistent information extracted from the same input $\boldsymbol{x}$. This objective inherently promotes the maximization of mutual information $I(\boldsymbol{z}_1; \boldsymbol{z}_2)$.

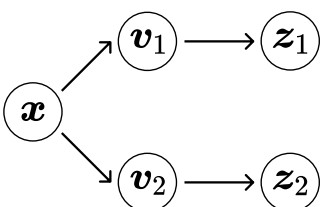

Due to the data processing inequality (Beaudry & Renner, 2012), maximization of the mutual information $I(\boldsymbol{z}_1; \boldsymbol{z}_2)$ serves as a lower bound for $I(\boldsymbol{z}_1; \boldsymbol{v}_2)$. As a result, this maximizes the mutual information between the learned representation and the alternate view, $I(\boldsymbol{z}_1; \boldsymbol{v}_2)$ (Tsai et al., 2021).

Figure 2: Graphical model for contrastive learning.

**Definition 1.** *(Sufficient representation for contrastive learning) A representation $\boldsymbol{z}_1^{suf}$ is considered sufficient for $\boldsymbol{v}_2$ if and only if $I(\boldsymbol{z}_1^{suf}; \boldsymbol{v}_2) = I(\boldsymbol{v}_1; \boldsymbol{v}_2)$*

This definition implies that a sufficient representation $\boldsymbol{z}_1^{suf}$ retains all the information that $\boldsymbol{v}_1$ contains about $\boldsymbol{v}_2$ (Wang et al., 2022a).

**Definition 2.** *(Minimal sufficient representation) A minimal sufficient representation $\boldsymbol{z}_1^{min}$ is considered minimal sufficient for $\boldsymbol{v}_2$ if and only if $I(\boldsymbol{z}_1^{min}; \boldsymbol{v}_1 | \boldsymbol{v}_2) = 0$, for all sufficient representations.*

The *superfluous information* refers to the information that is not shared between the two views, and it can be represented as conditional mutual information $I(\boldsymbol{z}_1; \boldsymbol{v}_1 | \boldsymbol{v}_2)$. A minimal sufficient representation $\boldsymbol{z}_1^{min}$ retains the least amount of this superfluous information for all sufficient representations. In multi-view information bottleneck (MVIB) research, minimal sufficient representations can be obtained by minimizing the superfluous information while maximizing the alignment between different views (Federici et al., 2020):

$$\mathcal{L}_{\text{MVIB}}(\phi) = \lambda I(\boldsymbol{z}_1; \boldsymbol{v}_1 | \boldsymbol{v}_2) - I(\boldsymbol{z}_1; \boldsymbol{v}_2), \tag{1}$$

where $\phi$ is the model parameter and $\lambda$ is a weighting constant. This learning approach has been shown in previous studies to facilitate more robust representation learning (Wan et al., 2021; Wen et al., 2024).

## 4 METHOD

### 4.1 PROBLEM FORMULATION AND NOTATIONS

We define the domain factor $D$ as the set of variables contributing to variability in biosignals across different individuals, including but not limited to factors such as age, gender, and pathological conditions. Let us denote the several domains as $\mathcal{D}_m := (\boldsymbol{x}_{k_m}, y_{k_m}, d_{k_m})_{k_m=1}^{|\mathcal{D}_m|}$, where $m \in \{1, 2, 3, \cdots, M\}$ denotes the $m$-th domain, $M$ is the number of domains. Here, $\boldsymbol{x}_{k_m}$ represents the physiological signal of $k$-th sample in $m$-th domain, $y_{k_m}$ is the corresponding sleep stage label, and $d_{k_m}$ is the domain label. A $k_m$-th sequence composed of $L$ signal samples is $\boldsymbol{X}_{k_m}^{L} = \{\boldsymbol{x}_{k_m-L}, \boldsymbol{x}_{k_m-L+1}, \cdots, \boldsymbol{x}_{k_m}\}$. The target domain $\mathcal{T}$ is defined as $\mathcal{D}_{m=T}$ and the source domain $\mathcal{S}$ is defined as $\mathcal{D}_{m \neq T}$, where $T$ represents the index set corresponding to the target subjects. The goal of our domain generalization in the sleep staging task is to learn the mapping function $g : \mathcal{X} \to \mathcal{Y}$ that can accurately predict the sleep stage ($y_{k_m}$) given a sequence of signals ($\boldsymbol{X}_{k_m}^{L}$) on unseen target domain $\mathcal{T}$, using only data from the source domains $\mathcal{S}$.

In the pre-training, we leverage the contrastive learning framework and randomly sampled $N$ instances set $\{\boldsymbol{x}_l, y_l, d_l\}_{l=1,\ldots,N}$ from the source domain. Each instance $\boldsymbol{x}_l$ is augmented to two views $\boldsymbol{v}_l, \boldsymbol{v}_{l+N}$, following the procedure outlined in SleePyCo (Lee et al., 2024). In a batch with multiple views, let $i \in B := \{1, \cdots, 2N\}$ be the index of the augmented sample, and $A(i) := B \setminus \{i\}$ be all index excluding $i$, where $\setminus$ indicates a set difference operator. The normalized feature from projection head of a sample $\boldsymbol{x}_i$ is denoted as $\boldsymbol{z}_i$.

(a) Multi-scale minimal sufficient representation learning

(b) Sleep staging

(c) Feature distribution

Figure 3: Overview of MSMS. Our method consists of two stages: (a) Multi-scale minimal sufficient representation learning and (b) Sleep staging. In (a), multi-scale features capturing diverse frequency and temporal information are projected into a shared feature space. Without conditional entropy $H(\boldsymbol{z}|d)$, features cluster by class but show domain misalignment. By maximizing $H(\boldsymbol{z}|d)$, the feature space expands, aligning domain distributions and improving domain-invariant representation, as illustrated in (c). In (b), the encoder is frozen, and the extracted multi-scale features are fed into a transformer to produce level-specific predictions. These are aggregated using argmax to determine the final sleep stage classification, following (Lee et al., 2024).

## 4.2 OVERALL FRAMEWORK

We adopt a CNN-based encoder and transformer-based sequential classifier to predict sleep stages using multi-scale features, following the approach proposed in SleePyCo (Lee et al., 2024). This method, which employs supervised learning for representation learning and incorporates multi-scale features for sleep staging, provides a robust and reliable baseline for our study.

First, the encoder is trained to extract multi-scale domain-invariant features by optimizing the objective in Eq. (11), as illustrated in Figure 3. Subsequently, the encoder is frozen, and the extracted multi-scale features are individually fed into the transformer. This process produces level-specific predictions, which are aggregated by taking the argmax values to determine the final sleep stage classification. Further implementation details are provided in Appendix B.6.

## 4.3 MINIMAL SUFFICIENT REPRESENTATION LEARNING

In contrastive learning-based DG, the feature space is typically encouraged to become more domain-invariant by increasing the similarity of samples belonging to the same class across various domains. However, while these methods may provide a sufficient presentation, they do not necessarily ensure the learning of a minimal sufficient representation. As a result, domain-relevant information that is not shared between different domains often remains within superfluous information, thereby making it insufficient to achieve domain-invariant features. We posit that minimal sufficient learning is more effective in obtaining domain-invariant features compared to contrastive learning-based approaches.

**Theorem 1.** *The sufficient representation $\boldsymbol{z}_1^{suf}$ contains more domain-relevant information than the minimal sufficient representation $\boldsymbol{z}_1^{min}$ (proof in Appendix A).*

$$I(\boldsymbol{z}_1^{suf}; d_1) \geq I(\boldsymbol{z}_1^{min}; d_1), \tag{2}$$

where $d$ refers to the domain label of $\boldsymbol{x}$ and $\boldsymbol{z}_1$, $\boldsymbol{z}_2$ is the normalized and projected outputs of two augmented views of $\boldsymbol{x}$.

Intuitively, this theorem holds because the superfluous information $I(\boldsymbol{z}_1; \boldsymbol{v}_1|\boldsymbol{v}_2)$ often encompasses domain-relevant information contained in $\boldsymbol{z}_1$. By minimizing this superfluous information, we can extract more domain-invariant features, which is crucial for domain generalization.

To formalize this intuition for the supervised setting, let $P(i) := \{p \in A(i) \mid y_p = y_i\}$ denote the set of indices for positive pairs. We can learn a minimal sufficient representation by reducing the

superfluous information $I(\boldsymbol{z}_i; \boldsymbol{v}_i|\boldsymbol{v}_p)$. Additionally, we minimize the domain-relevant information $I(\boldsymbol{z}_i; d_i)$ to effectively obtain domain-invariant features. Using the Lagrangian multiplier method, we can derive the following equation:

$$\mathcal{L}(\phi) = \lambda_1 I(\boldsymbol{z}_i; d_i) + \lambda_2 I(\boldsymbol{z}_i; \boldsymbol{v}_i|\boldsymbol{v}_p) - I(\boldsymbol{z}_i; \boldsymbol{v}_p), \tag{3}$$

where $\phi$ refer to model parameter, $\lambda_1$ and $\lambda_2$ are the Lagrangian multiplier. This loss function can be seen as an extension of the multi-view information bottleneck objectives in Eq. (1) to incorporate domain generalization and a supervised manner.

Mutual information is notoriously challenging to compute directly, particularly due to the requirement of estimating high-dimensional probability distributions. Recent advances (Wen et al., 2024) have addressed this challenge by approximating mutual information using the von Mises-Fisher (vMF) distribution, which is well-suited for modeling data constrained to a hypersphere. To leverage this approximation, we first express mutual information in terms of entropy. The Eq. (3) can be simplified by reducing the number of Lagrangian multiplier for computational convenience and reformulated in terms of entropy as follows (see Appendix B.1):

$$\mathcal{L}(\phi) = (\lambda + 1)H(\boldsymbol{z}_i|\boldsymbol{v}_p) - H(\boldsymbol{z}_i|d_i), \tag{4}$$

where $\lambda$ is the Lagrangian multiplier. Since the joint distribution $p(\boldsymbol{z}_i, \boldsymbol{v}_p)$ is unknown, directly calculating the conditional entropy $H(\boldsymbol{z}_i|\boldsymbol{v_p})$ becomes intractable. Therefore, we employ a variational approximation $q_\phi(\boldsymbol{z}_i, \boldsymbol{v}_p)$ and derive the upper bound:

$$H(\boldsymbol{z}_i|\boldsymbol{v}_p) = -\mathbb{E}_{p(\boldsymbol{z}_i, \boldsymbol{v}_p)}[\log p(\boldsymbol{z}_i|\boldsymbol{v}_p)] \tag{5}$$

$$= -\mathbb{E}_{p(\boldsymbol{z}_i, \boldsymbol{v}_p)}[\log q_\phi(\boldsymbol{z}_i|\boldsymbol{v}_p)] - D_{\mathrm{KL}}(p(\boldsymbol{z}_i|\boldsymbol{v}_p)||q_\phi(\boldsymbol{z}_i|\boldsymbol{v}_p)) \tag{6}$$

$$\leq -\mathbb{E}_{p(\boldsymbol{z}_i, \boldsymbol{v}_p)}[\log q_\phi(\boldsymbol{z}_i|\boldsymbol{v}_p)]. \tag{7}$$

Hence, minimization of Eq. (4) can be achieved through the following objective:

$$\bar{\mathcal{L}}(\phi) = -(\lambda + 1)\mathbb{E}_{p(\boldsymbol{z}_i, \boldsymbol{v}_p)}[\log q_\phi(\boldsymbol{z}_i|\boldsymbol{v}_p)] - H(\boldsymbol{z}_i|d_i). \tag{8}$$

To approximate $\mathbb{E}_{p(\boldsymbol{z}_i, \boldsymbol{v}_p)}[\log q_\phi(\boldsymbol{z}_i|\boldsymbol{v}_p)]$ and $H(\boldsymbol{z}_i|d_i)$, we utilize the von Mises-Fisher (vMF) distribution and Stein gradient estimation (Li & Turner, 2017). Consequently, we can optimize the Eq. (8) by minimize the following objective (see Appendix B.2 for comprehensive details):

$$\hat{\mathcal{L}}(\phi) = -\mathbb{E}_{p(\boldsymbol{z}_i, \boldsymbol{z}_p)}[\boldsymbol{z}_i \cdot \boldsymbol{z}_p] - \beta H(\boldsymbol{z}_i|d_i), \tag{9}$$

where $\beta$ is the balance factor.

However, the aforementioned objective lacks sufficient class discriminative power, as maximizing the conditional entropy $H(\boldsymbol{z}_i|d_i)$ tends to diffuse the feature space. To address this limitation, we introduce a negative pair term that pushes samples from different classes farther apart. This approach encourages the feature space to become more distinguishable by clustering samples of the same class, commonly utilized in contrastive learning. To ensure consistency within the contrastive learning framework, the cosine similarity is scaled by the temperature parameter $\tau$. This integrated objective can be expressed as follows (more details in Appendix B.4):

$$\tilde{\mathcal{L}}(\phi) = \sum_{i \in I} \frac{-1}{|P(i)|} \sum_{p \in P(i)} \log \frac{\exp(\boldsymbol{z}_i \cdot \boldsymbol{z}_p/\tau)}{\sum_{n \in N(i)} \exp(\boldsymbol{z}_i \cdot \boldsymbol{z}_n/\tau)} - \alpha H(\boldsymbol{z}_i|d_i), \tag{10}$$

where $N(i) := \{n \in A(i) \,|\, y_n \neq y_i\}$ is the set of indices of negative pairs for $i$-th instance in batch, $|P(i)|$ refer to cardinality of positive pair set and $\alpha$ is regularization parameter. $H(\boldsymbol{z}|d)$ can be alternatively expressed as $-\sum_{d_i=1}^{M} \mathbb{E}_{p(\boldsymbol{v}|d_i)}[\hat{\mathbf{G}}_m^{\mathrm{Stein}} \boldsymbol{z}]$ in gradient descent optimization, where $\hat{\mathbf{G}}_m^{\mathrm{Stein}}$ represents the score function derived using Stein gradient approximation (Li & Turner, 2017) for the $m$-th domain, similar to the approach in (Wen et al., 2024). This alternative is valid because its gradient, $-\sum_{d_i=1}^{M} \mathbb{E}_{p(\boldsymbol{v}|d_i)}[\hat{\mathbf{G}}_m^{\mathrm{Stein}} \nabla_\phi f_\phi(\boldsymbol{v}|d_i)]$, serves an approximation of $\nabla_\phi H(\boldsymbol{z}|d)$, as further detailed in Appendix B.2.

This objective can be viewed as an extension of traditional contrastive learning, incorporating a regularization term to facilitate domain generalization. Maximizing the conditional entropy $H(\boldsymbol{z}_i|d_i)$ prevents the clustering of samples from the same domain, thereby promoting the extraction of domain-invariant features. This limitation is particularly significant in sleep staging tasks, where multi-level features are essential for capturing distinct frequency characteristics.

### 4.4 Multi-scale Minimal Sufficient Representation Learning

While minimal sufficient learning at the higher level is crucial for mitigating domain gaps, this process carries the risk of inadvertently discarding essential information in the sub-level features due to the reduction of information. This limitation is critical in sleep stage tasks, where multi-level features capture distinct frequency characteristics. For example, slow-wave sleep (N3) is associated with low frequencies (0.5–2 Hz), captured by lower-level features, while Wake involves higher-frequency patterns (8–30 Hz), represented by higher-level features (Berry, 2014; Lee et al., 2024). Therefore, it is essential to ensure that feature information across multiple levels is preserved for accurate sleep stage classification.

To achieve this, we aim to employ minimal sufficient representation learning across multiple scales to effectively capture the diverse temporal and frequency characteristics present across different sleep stages. The objective for minimal sufficient representation learning in Eq. (10) can be extended to account for multi-scale features as follows:

$$\mathcal{L}_{\text{pre}}(\phi) = \sum_{j} \sum_{i \in I} \frac{-1}{|P(i)|} \sum_{p \in P(i)} \log \frac{\exp(\boldsymbol{z}_{i,j} \cdot \boldsymbol{z}_{p,j}/\tau)}{\sum_{n \in N(i)} \exp(\boldsymbol{z}_{i,j} \cdot \boldsymbol{z}_{n,j}/\tau)} - \alpha H(\boldsymbol{z}_{i,j}|d_i), \quad (11)$$

where $\boldsymbol{z}_{i,j}$ refers to the normalized feature from the $j$-th output of the encoder layer for the $i$-th instance. This objective ensures that the model captures not only various temporal and spectral information but also mitigates domain bias.

## 5 Experiment

### 5.1 Dataset

We evaluated the performance of our proposed method on two different sleep staging datasets: SleepEDF-20 (Kemp et al., 2000) and Montreal Archive of Sleep Studies (MASS) (O'reilly et al., 2014). The sleepEDF-20 dataset comprises PSG recordings from 20 subjects aged from 25 to 34. MASS contains PSG recordings from 62 subjects aged from 25 to 69. For the SleepEDF-20 dataset, we extracted a single-channel EEG (Fpz-Cz) sampled at 100Hz, and for the MASS dataset, we utilized the F4-LER channel, downsampled to 100Hz. For both SleepEDF-20 datasets, we combined the N3 and N4 stages into a single N3 stage. The class distribution of two datasets is in Appendix B.5. This process is a commonly used data preprocessing method in sleep staging, and we adhered to the settings of numerous previous studies to ensure a fair comparison (Seo et al., 2020; Phyo et al., 2022; Lee et al., 2024). In this study, we treat each subject as a separate domain, aligning with the common practice in sleep research. This approach accounts for the substantial inter-subject variability in physiological signals, as highlighted in prior works (Phan et al., 2021; Yang et al., 2023; Ko et al., 2024).

### 5.2 Implementations Details

The model was pre-trained with a batch size of 1024, an initial learning rate of $5 \times 10^{-4}$, and a weight decay of $1 \times 10^{-4}$ for the Adam optimizer. The temperature hyperparameter $\tau$ for the contrastive loss was set to 0.07, while the regularization parameter $\alpha$ was set to 0.001. We extracted features from the last two layers ($j = 4, 5$) of the encoder to align the multi-scale feature. The sleep staging process follows the same architecture as the transformer-based classifier utilizing multi-scale features ($j = 3,4,5$), as proposed in SleePyCo. For sleep staging, the pre-trained encoder was frozen, and only the classifier was trained, with the sequence length set to $L = 10$.

We employed the widely adopted k-fold cross-validation protocol to evaluate the performance of domain generalization. For each fold, we designated specific unseen subjects as the test set and repeated the experiment, ensuring that each subject was included in the test set exactly once. For the SleepEDF-20 dataset (k = 20), we partitioned the data into training, validation, and test sets with a ratio of 15:4:1, respectively. For the MASS dataset (k = 31), we used a ratio of 45:15:2 for training, validation, and test sets. All experiments were conducted on a server equipped with an NVIDIA RTX A6000 D6 48GB GPU.

Table 1: Performance comparison between Ours and sleep staging SOTA methods, and DG approaches for sleep staging on SleepEDF-20 and MASS dataset. We evaluated performance using three metrics: accuracy (ACC), macro-averaged F1 score (F1), and Cohen's Kappa ($\kappa$).

| Datasets | Method | Overall metrics | | |
|---|---|---|---|---|
| | | ACC (%) | F1 (%) | $\kappa$ |
| SleepEDF-20 | XSleepNet (Phan et al., 2021) | 86.3 | 80.6 | 0.813 |
| | Dream (Lee et al., 2022) | 83.9 | 75.7 | 0.770 |
| | Regularized SeqSleepNet (Phan et al., 2023) | 86.2 | 79.3 | 0.811 |
| | SleePyCo (Base) (Lee et al., 2024) | 86.2 | 80.6 | 0.812 |
| | ERM (Vapnik, 1998) | 84.0 | 76.9 | 0.777 |
| | IRM (Arjovsky et al., 2019) | 84.2 | 77.4 | 0.783 |
| | PCL (Yao et al., 2022) | 86.0 | 80.1 | 0.809 |
| | SleepDG (Wang et al., 2024a) | 84.8 | 78.4 | 0.792 |
| | COMET (Wang et al., 2024b) | 84.8 | 79.1 | 0.792 |
| | MSMS (Ours) | **86.7** | **81.1** | **0.818** |
| MASS | IITNet (Seo et al., 2020) | 86.3 | 80.5 | 0.794 |
| | ProductGraph (Einizade et al., 2023) | 86.7 | 81.8 | 0.802 |
| | SleepMG (Ma et al., 2024) | 86.6 | 81.7 | 0.802 |
| | SleePyCo (Base) (Lee et al., 2024) | 88.0 | 82.8 | 0.821 |
| | ERM (Vapnik, 1998) | 86.5 | 81.4 | 0.792 |
| | IRM (Arjovsky et al., 2019) | 87.7 | 82.5 | 0.817 |
| | PCL (Yao et al., 2022) | 87.9 | 82.9 | 0.819 |
| | SleepDG (Wang et al., 2024a) | 85.1 | 77.9 | 0.778 |
| | COMET (Wang et al., 2024b) | 87.5 | 82.7 | 0.815 |
| | MSMS (Ours) | **88.3** | **83.6** | **0.826** |

## 5.3 RESULTS

We conducted a comprehensive evaluation in comparison to state-of-the-art methods for sleep staging, as well as various domain generalization techniques, including ERM (empirical risk minimization) (Vapnik, 1998), IRM (minimizing risk across different environments) (Arjovsky et al., 2019), PCL (a proxy-based contrastive learning approach) (Yao et al., 2022), COMET (hierarchical contrastive learning in medical time series) (Wang et al., 2024b), and SleepDG (distribution matching of both global and local sleep sequences) (Wang et al., 2024a). All DG approaches, except for SleepDG, were trained using the SleePyCo backbone. The comparison was carried out using multiple metrics, including accuracy (ACC), macro-averaged F1 score (F1), and Cohen's Kappa ($\kappa$). As shown in Table 1 , our method demonstrated superior performance across both benchmark datasets, SleepEDF-20 and MASS.

Specifically, for the SleepEDF-20 dataset, our approach achieved an accuracy of 86.7%, an F1 score of 81.1%, and a $\kappa$ of 0.818. Similarly, for the MASS dataset, our method yielded competitive results with an accuracy of 88.3%, an F1 score of 83.6%, and a $\kappa$ of 0.826. We conducted statistical t-testing between our method and the baseline, calculating a p-value ($P < 0.001$) on both SleepEDF-20 and MASS datasets. The experimental results demonstrate the effectiveness of our proposed method over the baseline model, SleePyCo (Base), which employed supervised contrastive learning (SCL). Additionally, our approach outperforms other competitive contrastive learning-based domain generalization methods, such as PCL and COMET. This illustrates that our method can more effectively handle domain shifts, leading to better generalization on challenging sleep staging task.

## 5.4 ABLATION STUDIES

**Effect of Multi-Scale and Minimal Sufficient Representation Learning on Model Performance.**
To demonstrate the validity of MSMS, we performed ablation experiments to evaluate the impact of multi-scale features and minimal sufficient representation learning, as shown in Figure 4. Our method consistently performs well across both datasets and all three evaluation metrics. However,

Our method using SCL without minimal sufficient learning (Ours (w/o minimal)) did not result in performance improvements over standard SCL, which may be due to the accumulation of superfluous information from earlier layers. Conversely, an important observation is that Ours (w/o multi) underperforms compared to the baseline on SleepEDF-20, likely due to the reduced information in lower-level features caused by the pruning of high-level feature information. This highlights the necessity of combining minimal sufficient representation learning with multi-scale learning to effectively preserve meaningful information in low-level features.

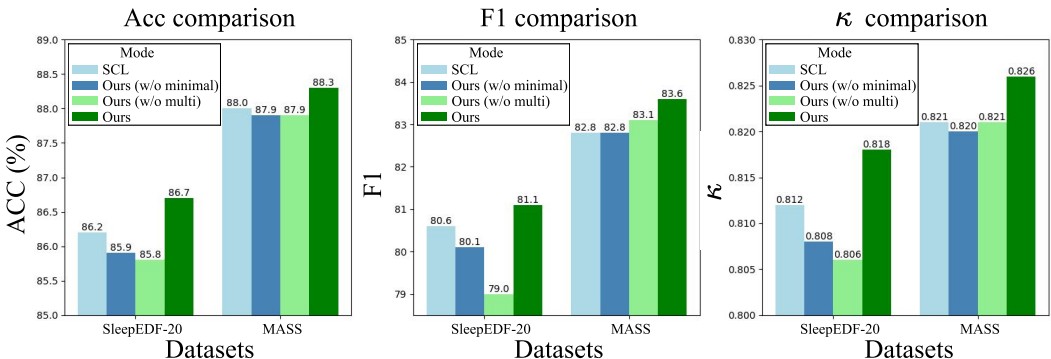

Figure 4: Ablation study results comparing the performance of different models on SleepEDF-20 and MASS datasets. SCL (w/ multi) refers to supervised contrastive learning with multi-scale learning, while Ours (w/o multi) refers to our proposed model without multi-scale learning.

**Exploring Optimal Feature Alignment Levels** We conducted ablation studies to determine which level of features should be aligned for optimal performance. Among the five encoder layers, we used the output features from the final layer (high-level), the fourth layer (middle-level), and the third layer (low-level). The results of this analysis are presented in Table 2. Our findings reveal that aligning only high-level features leads to a decrease in performance, whereas including other-level feature alignment results in significant performance improvement. The observed decline is likely due to the loss of information from previous feature layers by reducing superfluous information. Aligning only high-level features led to a decline in the model's performance across other sleep stages besides the Wake stage. This effect can be attributed to the high-level features effectively capturing high-frequency information, such as the distinctive beta rhythm (13–30 Hz) commonly observed in the Wake stage. This result underscores the necessity of employing multi-scale learning to preserve information across feature hierarchies.

**Analysis of regularization parameter $\alpha$.** We conducted ablation studies to evaluate the influence of the regularization parameter $\alpha$ on model performance, as illustrated in Figure 5. The optimal performance was achieved at $\alpha = 0.001$, indicating that appropriate regularization plays a crucial role in enhancing domain generalization. In contrast, larger values of $\alpha$ led to an overemphasis on $H(z_i|d_i)$, resulting in a failure to capture meaningful features and a subsequent decline in performance. These results highlight the significance of carefully balancing regularization to ensure the model retains class-relevant information while mitigating the influence of domain biases.

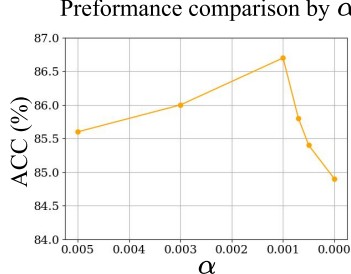

Figure 5: Performance comparison across varying the $\alpha$ on SleepEDF-20.

### 5.5 ANALYSIS

**t-SNE visualization.** We performed a feature visualization to further demonstrate the effectiveness of our method, as illustrated in Figure 6. The t-SNE visualizations show distributions of features between source (green) and target (orange) domains. For effective visualization in SleepEDF-20,

Table 2: Performance comparison of feature alignment at different levels on SleepEDF-20 datasets. We utilized the features extracted from the last encoder layer (high-level), the fourth layer (middle-level), and the third layer (low-level).

| High | Middle | Low | ACC (%) | F1 (%) | $\kappa$ | W | N1 | N2 | N3 | REM |
|---|---|---|---|---|---|---|---|---|---|---|
| ✓ | | | 85.8 | 79.0 | 0.806 | **93.2** | 43.8 | 88.3 | **88.0** | 81.9 |
| ✓ | | ✓ | 86.2 | 80.5 | 0.811 | 90.5 | 50.1 | 88.3 | 87.9 | 85.8 |
| ✓ | ✓ | | **86.7** | **81.1** | **0.818** | 91.9 | 51.2 | **89.0** | 87.3 | **86.3** |
| ✓ | ✓ | ✓ | 86.5 | 81.1 | 0.816 | 91.5 | **51.7** | 88.8 | **88.0** | 85.5 |

we selected subject 9, which exhibits significant variation, as the target for our analysis. The feature distribution in SCL exhibits misalignment between the source and target domains. In contrast, our MSMS approach achieves a much more aligned distribution between these domains, indicating that our method effectively generalizes unseen data well.

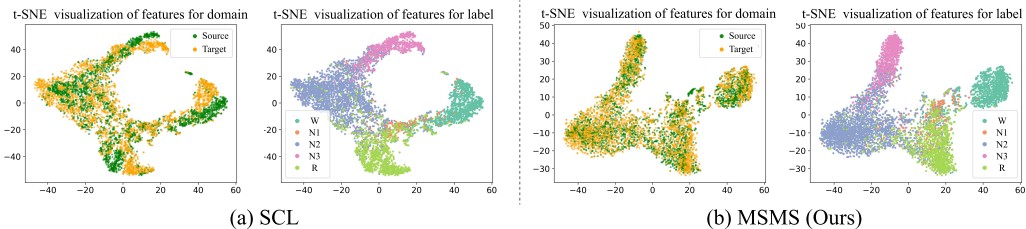

(a) SCL        (b) MSMS (Ours)

Figure 6: t-SNE visualization of features distribution on SleepEDF-20. We assigned subject 9 as the target (orange) and used the remaining 15 subjects, excluding the validation set, as the source (green) in this figure.

**Analysis of Superfluous and Domain-Relevant Information Correlations.** To assess whether our method effectively reduces superfluous information and captures domain-invariant features, we conducted an analysis comparing three different approaches, as illustrated in Figure 7. The information quantities at high-level features depicted in the figure were approximated using the vMF distribution, which is used in our method. Our method achieved the lowest quantities of superfluous information $I(z_i; v_i|v_p)$ and domain-relevant information $I(z_i|d_i)$, suggesting that our approach effectively minimizes both during training. Additionally, we observed a proportional relationship between superfluous and domain-relevant information, which supports Theorem 1—minimizing superfluous information leads to a reduction in domain-relevant information.

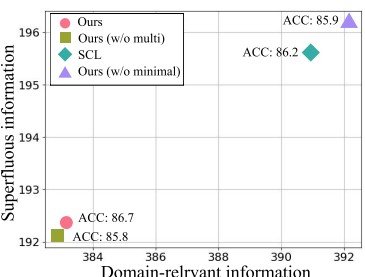

Figure 7: Visualization of correlation between the superfluous information and domain-relevant information.

## 6 CONCLUSION

In this work, we proposed a novel framework, Multi-Scale Minimal Sufficient representation learning (MSMS), which mitigates domain gaps by reducing superfluous information while simultaneously aligning multi-scale features to consider various temporal and spectral characteristics inherent in physiological signals. Extensive experiments conducted on publicly available sleep staging datasets demonstrate that our approach consistently outperforms SOTA techniques. These results highlight that our method ensures generalization to unseen domains. While this framework demonstrates promising results, future research will explore incorporating mechanisms to mitigate distribution shifts across datasets, thereby improving the framework's robustness and applicability in real-world scenarios.

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

## A  PROOF OF THEOREM 1

**Theorem** 1 The sufficient representation $z_1^{suf}$ contains more domain-relevant information than the minimal sufficient representation $z_1^{min}$.

**Proof:** First, recall that $z_1^{suf}$ is a sufficient representation of $v_1$ with respect to $v_2$, meaning: $I(z_1^{suf}; v_2) = I(v_1; v_2)$. We also know that $z_1^{min}$ is a minimal sufficient representation, which means it is derived from $z_1^{suf}$ and satisfies: $I(z_1^{min}; v_1 | v_2) = 0$. We begin by examining the mutual information between the sufficient representation $z_1^{suf}$ and the domain label $d$:

$$I(z_1^{suf}; d) = H(d) - H(d|z_1^{suf}) \tag{12}$$

$$= H(d) - H(d|z_1^{suf}, v_2) - I(d; v_2|z_1^{suf}) \tag{13}$$

$$= [H(d) - H(d|v_2)] + [H(d|v_2) - H(d|z_1^{suf}, v_2)] - I(d; v_2|z_1^{suf}) \tag{14}$$

$$= I(d; v_2) + I(z_1^{suf}; d|v_2) - I(d; v_2|z_1^{suf}) \tag{15}$$

$$\geq I(d; v_2) + I(z_1^{suf}; d|v_2) - I(d; v_2|z_1^{min}) \tag{16}$$

$$= I(z_1^{suf}; d|v_2) + I(z_1^{min}; d) \tag{17}$$

$$\geq I(z_1^{min}; d). \tag{18}$$

Here is the explanation for each step:

Eq. (12) Now, let's consider the joint distribution of $(z_1^{suf}, v_2, d)$. We can express the mutual information $I(z_1^{suf}; d)$ in terms of entropies: $I(z_1^{suf}; d) = H(d) - H(d|z_1^{suf})$.

Eq. (13) We can further decompose this using the chain rule of entropy: $I(z_1^{suf}; d) = H(d) - H(d|z_1^{suf}, v_2) - I(d; v_2|z_1^{suf})$.

Eq. (14) Rearranging this equation: $I(z_1^{suf}; d) = [H(d) - H(d|v_2)] + [H(d|v_2) - H(d|z_1^{suf}, v_2)] - I(d; v_2|z_1^{suf})$.

Eq. (15) Recognizing mutual information terms, we get Eq (15): $I(z_1^{suf}; d) = I(d; v_2) + I(z_1^{suf}; d|v_2) - I(d; v_2|z_1^{suf})$.

Eq. (16) Inequality Eq. (16) is due to the data processing inequality. Since $z_1^{min}$ is a function of $z_1^{suf}$, we have $I(d; v_2|z_1^{suf}) \leq I(d; v_2|z_1^{min})$.

Eq. (17) Eq. (17) uses the definition of minimal sufficient representation. For $z_1^{min}$, we have $I(z_1^{min}; d) = I(d; v_2) - I(d; v_2|z_1^{min})$.

Eq (18) Inequality Eq (18) holds because mutual information is non-negative, so $I(z_1^{suf}; d|v_2) \geq 0$.

Therefore, we have shown that $I(z_1^{suf}; d) \geq I(z_1^{min}; d)$, which means that the sufficient representation $z_1^{suf}$ contains more domain-relevant information than the minimal sufficient representation $z_1^{min}$.

## B  PROOF OF MINIMAL SUFFICIENT LEARNING METHOD

In this section, we provide the formal proof of the minimal sufficient learning method proposed in the paper.

### B.1  PROOF OF EQ. (4)

Eq. (4) : $\mathcal{L}(\phi) = (\lambda + 1)H(z_i|v_p) - H(z_i|d_i)$.

The superfluous information $I(z_i; v_i|v_p)$ can be decomposed as:

$$I(z_i; v_i|v_p) = H(z_i|v_p) - H(z_i|v_i, v_p) \tag{19}$$

$$= H(z_i|v_p), \tag{20}$$

where the conditional entropy $H(\boldsymbol{z}_i|\boldsymbol{v}_i, \boldsymbol{v}_p) = 0$ because $\boldsymbol{z}_i$ is determined given $\boldsymbol{v}_i$ (we used deterministic encoder). We can also decompose mutual information as:

$$I(\boldsymbol{z}_i; \boldsymbol{v}_p) = H(\boldsymbol{z}_i) - H(\boldsymbol{z}_i|\boldsymbol{v}_p) \tag{21}$$

$$I(\boldsymbol{z}_i; d_i) = H(\boldsymbol{z}_i) - H(\boldsymbol{z}_i|d_i). \tag{22}$$

Based on the above derivations and Eq. (3), we finally obtain the general objective below:

$$\mathcal{L}(\phi) = \lambda_1 I(\boldsymbol{z}_i; d_i) + \lambda_2 I(\boldsymbol{z}_i; \boldsymbol{v}_i|\boldsymbol{v}_p) - I(\boldsymbol{z}_i; \boldsymbol{v}_p) \tag{23}$$

$$= \lambda_1(H(\boldsymbol{z}_i) - H(\boldsymbol{z}_i|d_i)) + \lambda_2(H(\boldsymbol{z}_i|\boldsymbol{v}_p)) - H(\boldsymbol{z}_i) + H(\boldsymbol{z}_i|\boldsymbol{v}_p) \tag{24}$$

$$= (\lambda_2 + 1)(H(\boldsymbol{z}_i|\boldsymbol{v}_p)) + (\lambda_1 - 1)H(\boldsymbol{z}_i) - \lambda_1 H(\boldsymbol{z}_i|d_i). \tag{25}$$

For the sake of computational convenience and to simplify the search for optimized parameters, we set $\lambda_1 = 1$. Consequently, we can obtain the objective as follows:

$$(\lambda + 1)H(\boldsymbol{z_i}|\boldsymbol{v_p}) - H(\boldsymbol{z_i}|d_i), \tag{26}$$

where $\lambda_2$ is redefined as $\lambda$.

## B.2 PROOF OF EQ. (9)

Proof of Eq. (9): $\mathcal{L}(\phi) = -\mathbb{E}_{p(\boldsymbol{z}_i, \boldsymbol{z}_p)}[\boldsymbol{z}_i, \boldsymbol{z}_p] - \beta H(\boldsymbol{z}_i|d_i)$.

The von Mises–Fisher (vMF) is the common distribution of the hypersphere space:

$$p(\boldsymbol{x}; \boldsymbol{\mu}, \kappa) = C_n(\kappa)\exp(\kappa\boldsymbol{\mu} \cdot \boldsymbol{x}), \tag{27}$$

$$C_n(\kappa) = \frac{\kappa^{n/2-1}}{(2\pi)^{n/2}I_{n/2-1}(\kappa)}, \tag{28}$$

where $\boldsymbol{\mu}$ is the mean direction, $\kappa$ denotes the concentration parameter of the vMF distribution, and $I_n$ denotes the modified Bessel function of the first kind at order $n$.

The representation $\boldsymbol{z}$ is $\ell_2$-normalized in the hypersphere space. Hence, The variational distribution $q_\phi(\boldsymbol{z}_i|\boldsymbol{v}_p)$ can be adequately approximated by the vMF distribution as, similar to (Wen et al., 2024):

$$q_\phi(\boldsymbol{z}_i|\boldsymbol{v}_p) = C_n(\kappa)\exp(\kappa\boldsymbol{z}_p \cdot \boldsymbol{z}_i). \tag{29}$$

We assume that $\kappa$ is constant and use $\boldsymbol{z}_p$ as $\boldsymbol{\mu}$. Hence, Eq. (7) can be reformulated as follows:

$$H(\boldsymbol{z}_i|\boldsymbol{v}_p) \leq -\mathbb{E}_{p(\mathbf{z}_i, \mathbf{v}_p)}[\kappa\boldsymbol{z}_p \cdot \boldsymbol{z}_i] - \log C_n(\kappa). \tag{30}$$

Eq. (8) can be expressed as follows:

$$\bar{\mathcal{L}}(\phi) = -\mathbb{E}_{p(\mathbf{z}_i, \mathbf{v}_p)}[\boldsymbol{z}_p \cdot \boldsymbol{z}_i] - \beta H(\boldsymbol{z}_i|d_i), \tag{31}$$

where $\beta = \frac{1}{(\lambda+1)\kappa}$ is the balance factor.

## B.3 COMPUTATION OF ENTROPY

The conditional entropy term $H(\boldsymbol{z}_i|d_i)$ in Eq. (31) is anticipated to be maximized during model training. Thus, we maximize the $H(\boldsymbol{z}_i|d_i)$ to use stein gradient estimation (Li & Turner, 2017). We follow the derivation from (Wen et al., 2024), with the key difference being that it is conditioned on the given parameter $d_i$.

The gradient of $H(\boldsymbol{z}_i|d)$ w.r.t. $\phi$ can be decomposed as:

$$\nabla_{\boldsymbol{\phi}}H(\boldsymbol{z}|d_i) = -\nabla_{\boldsymbol{\phi}}\mathbb{E}_{q_\phi(\boldsymbol{z}, d_i)}[\log q(\boldsymbol{z}|d_i)] - \mathbb{E}_{q(\boldsymbol{z}, d_i)}[\nabla_\phi \log q_\phi(\boldsymbol{z}|d_i)], \tag{32}$$

where $q(\boldsymbol{z}, d)$ without the subscript $\phi$ means the gradient of computation is irrelevant to $\phi$. The second term can be further decomposed as:

$$\mathbb{E}_{q(\boldsymbol{z}, d_i)}[\nabla_\phi \log q_\phi(\boldsymbol{z}|d_i)] = \mathbb{E}_{q(\boldsymbol{z})}\left[\nabla_\phi q_\phi(\boldsymbol{z}|d_i) \times \frac{1}{q(\boldsymbol{z}|d_i)}\right] \tag{33}$$

$$= \int \nabla_\phi q_\phi(\boldsymbol{z}|d_i)d\boldsymbol{z} = \nabla_\phi \int q_\phi(\boldsymbol{z}|d_i)d\boldsymbol{z} = 0. \tag{34}$$

Hence we have

$$\nabla_\phi H(\boldsymbol{z}|d_i) = -\nabla_\phi \mathbb{E}_{q_\phi(\boldsymbol{z}, d_i)}[\log q(\boldsymbol{z}|d_i)]. \tag{35}$$

We adopt the reparameterization trick to address non-differentiable $H(\boldsymbol{z}|d_i)$ w.r.t $\phi$. We introduce the deterministic function $f_\phi$ and any joint distribution $p(\cdot)$ that is independent to model parameter $\phi$:

$$\boldsymbol{z} = f_\phi(\boldsymbol{v}|d_i) \quad \text{with} \quad \boldsymbol{v} \sim p(\boldsymbol{v}, d_i). \tag{36}$$

The conditional entropy gradient estimator is eventually derived as follows:

$$\nabla_\phi H(\boldsymbol{z}|d_i) = -\nabla_\phi \mathbb{E}_{q_\phi(\boldsymbol{z}, d_i)}[\log q(\boldsymbol{z}|d_i)] = -\mathbb{E}_{p(\boldsymbol{v}, d_i)}[\nabla_\phi \log q(f_\phi(\boldsymbol{v}|d_i))] \tag{37}$$

$$= -\mathbb{E}_{p(\boldsymbol{v}, d_i)}[\nabla_{\boldsymbol{z}} \log q(\boldsymbol{z}|d_i) \nabla_\phi f_\phi(\boldsymbol{v}|d_i)], \tag{38}$$

where $\nabla_{\boldsymbol{z}} \log q(\boldsymbol{z}|d)$ is the score function. $\nabla_\phi f_\phi(\boldsymbol{v}|d)$ can be obtained by direct back-propagation.

We use Stein gradient estimation (Li & Turner, 2017) to approximate the score function $\nabla_{\boldsymbol{z}} \log q(\boldsymbol{z}|d_i)$. Let $\boldsymbol{z}$ be supported on $\mathcal{Z} \subseteq \mathbb{R}^{d'}$, where $d'$ represents dimensionality of the feature space. Define $\boldsymbol{h}(\boldsymbol{z}) = [h_1(\boldsymbol{z}), h_2(\boldsymbol{z}), \dots, h_{d'}(\boldsymbol{z})]^T$ as a $d'$-dimensional differentiable vector function, satisfying the following boundary condition:

$$q(\boldsymbol{z})\boldsymbol{h}(\boldsymbol{z}) = \boldsymbol{0} \,, \forall \boldsymbol{z} \in \partial\mathcal{Z} \text{ if } \mathcal{Z} \text{ is compact, or } \lim_{\boldsymbol{z} \to \infty} q(\boldsymbol{z})\boldsymbol{h}(\boldsymbol{z}) = \boldsymbol{0} \text{ if } \mathcal{Z} = \mathbb{R}^{d'} \tag{39}$$

Then the following Stein's identity can be derived through the integration of parts:

$$\mathbb{E}_q \left[ \boldsymbol{h}(\boldsymbol{z})[\nabla_{\boldsymbol{z}} \log q(\boldsymbol{z})]^T + \nabla_{\boldsymbol{z}} \boldsymbol{h}(\boldsymbol{z}) \right] = \boldsymbol{0}, \tag{40}$$

The expectation in Eq. (40) can be estimated using the Monte Carlo method. Specifically, let $\boldsymbol{z}^{1:M'}$ represent $M'$ independent and identically distributed (i.i.d.) samples drawn from $q(\boldsymbol{z})$. Monte Carlo sampling shows:

$$-\frac{1}{M'}\boldsymbol{H}\boldsymbol{G} \approx \overline{\nabla_{\boldsymbol{z}}\boldsymbol{h}} \tag{41}$$

where $\mathbf{H} = \left[ \boldsymbol{h}\left(\boldsymbol{z}^1\right), \cdots, \boldsymbol{h}\left(\boldsymbol{z}^{M'}\right) \right] \in \mathbb{R}^{d'' \times M'}$, $\mathbf{G} = \left[ \nabla_{\boldsymbol{z}^1} \log q\left(\boldsymbol{z}^1\right), \cdots, \nabla_{\boldsymbol{z}^{M'}} \log q\left(\boldsymbol{z}^{M'}\right) \right]^T$ $\in \mathbb{R}^{M' \times d'}$, $\nabla_{\boldsymbol{z}^m} \boldsymbol{h}\left(\boldsymbol{z}^m\right) = [\nabla_{\boldsymbol{z}^m} h_1\left(\boldsymbol{z}^m\right), \dots, \nabla_{\boldsymbol{z}^m} h_{d'}\left(\boldsymbol{z}^m\right)]^T \in \mathbb{R}^{d'' \times d'}$, and $\overline{\nabla_{\boldsymbol{z}}\boldsymbol{h}} = \frac{1}{M'} \sum_{m=1}^{M'} \nabla_{\boldsymbol{z}^m} \boldsymbol{h}\left(\boldsymbol{z}^m\right) \in \mathbb{R}^{d'' \times d'}$ This leads to the following ridge regression formulation:

$$\underset{\hat{\mathbf{G}} \in \mathbb{R}^{M' \times d'}}{\arg\min} \left\| \overline{\nabla_{\boldsymbol{z}}\boldsymbol{h}} + \frac{1}{M}\mathbf{H}\hat{\mathbf{G}} \right\|_F^2 + \frac{\eta}{M^2}\|\hat{\mathbf{G}}\|_F^2, \tag{42}$$

where $\eta \leq 0$ serves as the regularization coefficient. An analytic solution of Eq. (42) is

$$\hat{\mathbf{G}}^{\text{Stein}} = -M'(\mathbf{K} + \eta\mathbf{I})^{-1}\mathbf{H}^T\overline{\nabla_{\boldsymbol{z}}\boldsymbol{h}}, \tag{43}$$

where $\mathbf{K} = \mathbf{H}^T\mathbf{H}$. Similar to (Wen et al., 2024), we express $\mathbf{K}_{i,j} = k(\boldsymbol{z}^i, \boldsymbol{z}^j)$, and establish $(\mathbf{H}^T\overline{\nabla_{\boldsymbol{z}}\boldsymbol{h}})_{i,j} = \frac{1}{M'}\nabla_{\boldsymbol{z}_j^m} k(\boldsymbol{z}^i, \boldsymbol{z}^j)$. We adopt the von Mises-Fhiser kernel defined as $k(\boldsymbol{z}, \boldsymbol{z}') = \exp\left(\frac{\boldsymbol{z}^T\boldsymbol{z}'}{\triangle}\right)$ to compute the $\hat{\mathbf{G}}^{\text{Stein}}$.

We approximate the score function $\nabla_{\boldsymbol{z}} \log q(\boldsymbol{z})$ as $\hat{\mathbf{G}}^{\text{Stein}}$. Based on this approximation, the entropy gradient estimator is formulated as:

$$\nabla_\phi H(\boldsymbol{z}) = -\mathbb{E}_{p(\boldsymbol{v})}[\nabla_{\boldsymbol{z}} \log q(\boldsymbol{z})\nabla_\phi f_\phi(\boldsymbol{v})] \tag{44}$$

$$\approx -\mathbb{E}_{p(\boldsymbol{v})}[\hat{\mathbf{G}}^{\text{Stein}}\nabla_\phi f_\phi(\boldsymbol{v})] \tag{45}$$

$$\nabla_\phi H(\boldsymbol{z}|d) = -\sum_{d_i=1}^{M} \mathbb{E}_{p(\boldsymbol{v}|d_i)}[\nabla_{\boldsymbol{z}} \log q(\boldsymbol{z}|d_i)\nabla_\phi f_\phi(\boldsymbol{v}|d_i)] \tag{46}$$

$$\approx -\sum_{d_i=1}^{M} \mathbb{E}_{p(\boldsymbol{v}|d_i)}[\hat{\mathbf{G}}_m^{\text{Stein}}\nabla_\phi f_\phi(\boldsymbol{v}|d_i)] \tag{47}$$

$$\tag{48}$$

where, $\hat{\mathbf{G}}_m^{\text{Stein}}$ represent the approximation of the score function $\nabla_{\boldsymbol{z}} \log q(\boldsymbol{z}|d_i)$ computed for the $m$-th domain.

$H(\boldsymbol{z}|d)$ can be alternatively represented as $-\sum_{d_i=1}^{M} \mathbb{E}_{p(\boldsymbol{v}|d_i)}[\hat{\mathbf{G}}_m^{\text{Stein}} \boldsymbol{z}]$ in decent gradient optimization. This is because its gradient, $-\sum_{d_i=1}^{M} \mathbb{E}_{p(\boldsymbol{v}|d_i)}[\hat{\mathbf{G}}_m^{\text{Stein}} \nabla_\phi f_\phi(\boldsymbol{v}|d_i)]$, provides an approximation of $\nabla_\phi H(\boldsymbol{z}|d)$, as described in Eq. (47).

### B.4 PROOF OF EQ. 10

$\mathbb{E}_{p(\boldsymbol{z}_i, \boldsymbol{z}_p)}[\boldsymbol{z}_i \cdot \boldsymbol{z}_p]$ can be decomposed using Monte Carlo approximation as:

$$\mathbb{E}_{p(\boldsymbol{z}_i, \boldsymbol{z}_p)}[\boldsymbol{z}_i \cdot \boldsymbol{z}_p] = \sum_{i \in I} \sum_{p \in P(i)} p(\boldsymbol{z}_p|\boldsymbol{z}_i) p(\boldsymbol{z}_i) \, \boldsymbol{z}_i \cdot \boldsymbol{z}_p \tag{49}$$

$$\approx \frac{1}{|I|} \sum_{i \in I} \sum_{p \in P(i)} \frac{1}{|P(i)|} \, \boldsymbol{z}_i \cdot \boldsymbol{z}_p, \tag{50}$$

$$\mathbb{E}_{p(\boldsymbol{z}_i, \boldsymbol{z}_p)}[\boldsymbol{z}_i \cdot \boldsymbol{z}_p/\tau] = \frac{1}{|I|} \sum_{i \in I} \sum_{p \in P(i)} \frac{1}{|P(i)|} \, \boldsymbol{z}_i \cdot \boldsymbol{z}_p/\tau. \tag{51}$$

Eq. (9) can rewrite as follows:

$$\hat{\mathcal{L}}(\phi)/\tau = -\frac{1}{|I|} \sum_{i \in I} \sum_{p \in P(i)} \frac{1}{|P(i)|} \, \boldsymbol{z}_i \cdot \boldsymbol{z}_p/\tau - \beta/\tau H(\boldsymbol{z}_i|d_i). \tag{52}$$

We also consider a set of negative pairs as follows:

$$\hat{\mathcal{L}}_{\text{w/neg}}(\phi)/\tau = -\frac{1}{|I|} \sum_{i \in I} \sum_{p \in P(i)} \frac{1}{|P(i)|} \, \boldsymbol{z}_i \cdot \boldsymbol{z}_p/\tau + \frac{1}{|I|} \sum_{i \in I} \log\Big( \sum_{n \in N(i)} \exp(\boldsymbol{z}_i \cdot \boldsymbol{z}_n/\tau) \Big)$$
$$- \beta/\tau H(\boldsymbol{z}_i|d_i) \tag{53}$$

$$= -\frac{1}{|I|} \sum_{i \in I} \frac{1}{|P(i)|} \sum_{p \in P(i)} \log \frac{\exp(\boldsymbol{z}_i \cdot \boldsymbol{z}_p/\tau)}{\sum_{n \in N(i)} \exp(\boldsymbol{z}_i \cdot \boldsymbol{z}_n/\tau)} - \beta/\tau H(\boldsymbol{z}_i|d_i). \tag{54}$$

We can minimize the Eq. (54) by minimizing the objective as follows:

$$\tilde{\mathcal{L}}(\phi) = -\sum_{i \in I} \frac{1}{|P(i)|} \sum_{p \in P(i)} \log \frac{\exp(\boldsymbol{z}_i \cdot \boldsymbol{z}_p/\tau)}{\sum_{n \in N(i)} \exp(\boldsymbol{z}_i \cdot \boldsymbol{z}_n/\tau)} - \alpha H(\boldsymbol{z}_i|d_i), \tag{55}$$

where $\alpha$ is regularization parameter.

### B.5 DATASET DESCRIPTION

We evaluated the effectiveness of our proposed approach using two distinct sleep staging datasets: SleepEDF-20 (Kemp et al., 2000) and the Montreal Archive of Sleep Studies (MASS) (O'reilly et al., 2014). A summary and distribution of classes for each dataset are presented in Table 3.

Table 3: Summary of sleep stage distribution for SleepEDF-20 and MASS datasets.

| Dataset | SleepEDF-20 | MASS |
|---------|-------------|------|
| W | 8285 (19.6 %) | 6231 (10.6 %) |
| N1 | 2804 (6.6 %) | 4814 (8.2 %) |
| N2 | 17799 (42.1 %) | 29777 (50.4 %) |
| N3 | 5703 (13.5 %) | 7653 (12.9 %) |
| REM | 7717 (18.2 %) | 10581 (17.9 %) |
| Total | 42308 | 59056 |

### B.6 Sleep Staging

We employ a transformer-based sequential classifier to predict sleep stages by leveraging multi-scale features, following the approach proposed in (Lee et al., 2024). The encoder, trained with our objective, is fixed, and the $k$-th sequence $\boldsymbol{X}_k^L$ is input to the encoder to extract the $j$-th level sequence features, denoted as $\boldsymbol{H}_j = \{\boldsymbol{h}_{k-L,j}, \boldsymbol{h}_{k-L+1,j}, \ldots, \boldsymbol{h}_{k,j}\}$, where $j \in \{3, 4, 5\}$. The attention sum of the transformer's hidden states for $\boldsymbol{H}_j$ is represented as $\tilde{\boldsymbol{h}}_j$. The prediction for the $j$-th level, $\boldsymbol{o}_j$, is obtained by passing $\tilde{\boldsymbol{h}}_j$ through a linear layer. The final predicted sleep stage, $\hat{y}_k$, is computed as follows:

$$\hat{y}_k = \arg\max \left( \sum_j \boldsymbol{o}_j \right)$$

### B.7 Distinguishing the roles of superfluous information and domain relevant information

To investigate the respective roles of minimizing superfluous $I(\boldsymbol{z}_i; d_i)$ and maximizing domain relevant information $I(\boldsymbol{z}_i; \boldsymbol{v}_i | \boldsymbol{v}_p)$, we conducted an ablation study on the SleepEDF-20 dataset. The results are presented in Table 4. The analysis reveals that training without $I(\boldsymbol{z}_i; \boldsymbol{v}_i | \boldsymbol{v}_p)$ leads to features that are less distinguishable across classes. This is likely due to the influence of minimization $I(\boldsymbol{z}_i; d_i)$, which, while suppressing domain-specific information, may inadvertently discard critical class-related information as well. These findings emphasize that if the loss for $I(\boldsymbol{z}_i; d_i)$ is to be used, it is essential to include a minimization of superfluous information term for $I(\boldsymbol{z}_i; \boldsymbol{v}_i | \boldsymbol{v}_p)$, which helps encode meaningful and relevant information within the features. On the other hand, when using only $I(\boldsymbol{z}_i; \boldsymbol{v}_i | \boldsymbol{v}_p)$ without $I(\boldsymbol{z}_i; d_i)$, the model's generalization ability is reduced. We will include this analysis and the corresponding table in the final version to further illustrate the effect of each term on feature representation and model performance.

Table 4: Performance comparison based on mutual information components.

| $I(\boldsymbol{z}_i; d_i)$ | $I(\boldsymbol{z}_i; \boldsymbol{v}_i \mid \boldsymbol{v}_p)$ | ACC (%) | F1 (%) | $\kappa$ |
|:---:|:---:|:---:|:---:|:---:|
| ✓ | | 79.1 | 78.1 | 0.712 |
| | ✓ | 86.2 | 80.3 | 0.809 |
| ✓ | ✓ | **86.7** | **81.1** | **0.818** |

### B.8 Analysis of $\lambda_1$ in Eq. (25)

In our initial experiments, we set $\lambda_1$ in Eq. (25), ignoring the influence of $H(\boldsymbol{z})$ for computational simplicity. To evaluate the impact of this design choice, we conducted experiments on the SleepEDF-20 dataset by varying the value of $\lambda_1$. The results of these experiments are presented in Table 5.

Table 5: Performance results for different values of $\lambda_1$.

| $\lambda_1$ | ACC (%) | F1 (%) | $\kappa$ |
|:---:|:---:|:---:|:---:|
| 0.1 | 85.8 | 79.7 | 0.806 |
| 0.5 | 86.0 | 80.3 | 0.809 |
| 0.7 | 85.9 | 80.1 | 0.807 |
| 1 | **86.7** | **81.1** | **0.818** |
| 1.5 | 86.3 | 80.4 | 0.811 |
| 2 | 86.2 | 80.2 | 0.810 |

When $\lambda_1$ is not fixed to 1, the objective function in Eq. (11) can be expressed as follows:

$$\mathcal{L}_{\text{pre}}(\phi) = \sum_j \sum_{i \in I} \frac{-1}{|P(i)|} \sum_{p \in P(i)} \log \frac{\exp(\boldsymbol{z}_{i,j} \cdot \boldsymbol{z}_{p,j} / \tau)}{\sum_{n \in N(i)} \exp(\boldsymbol{z}_{i,j} \cdot \boldsymbol{z}_{n,j} / \tau)} + \alpha(\lambda_1 - 1) H(\boldsymbol{z}_{i,j}) - \alpha \lambda_1 H(\boldsymbol{z}_{i,j} \mid d_i)$$

$$(56)$$

In the case where $\lambda_1 > 1$, the coefficient in front of $H(\boldsymbol{z}_{i,j})$ is positive, causing the model to attempt to minimize $H(\boldsymbol{z}_{i,j})$. Minimizing $H(\boldsymbol{z}_{i,j})$ reduces the amount of information contained in $\boldsymbol{z}$, which appears to hinder the learning process.

In the case where $\lambda_1 \leq 1$, the model simultaneously maximizes both $H(\boldsymbol{z}_{i,j})$ and $H(\boldsymbol{z}_{i,j}|d)$. While maximizing $H(\boldsymbol{z}_{i,j}|d)$ minimize the domain-relevant information $I(\boldsymbol{z}_{i,j}; d_i)$, maximizing $H(\boldsymbol{z}_{i,j})$ increases $I(\boldsymbol{z}_{i,j}; d_i)$, as $I(\boldsymbol{z}; d) = H(\boldsymbol{z}) - H(\boldsymbol{z}|d)$. Therefore, setting $\lambda_1 = 1$ allows the model to focus entirely on maximizing $H(\boldsymbol{z}_{i,j}|d)$, enabling the extraction of more domain-invariant features and improving the model's ability to generalize across domains.

