# OpenReview forum: "Multi-scale Minimal Sufficient Representation Learning for Domain Generalization in Sleep Staging"
_ICLR.cc/2025/Conference — ICLR 2025 Conference Withdrawn Submission_

### Official Review · Reviewer_4jps · 2024-10-24

**Soundness:** 3
**Presentation:** 3
**Contribution:** 2
**Rating:** 6
**Confidence:** 3

**Summary:**

This paper proposes a domain generalization method for sleep stage prediction, multi-scale minimal sufficient representation learning (MSMS).
The basic approach of MSMS is learning domain-invariant representation as existing domain generalization methods.
This paper argues that the representations learned by existing methods still contain domain-relevant information called superfluous information.
Thus, the proposed method explicitly incorporates the superfluous information into the objective.
Another idea of the proposed method is learning domain-invariant representation in the final encoder output space and intermediate feature spaces because capturing multi-scale characteristics of sleep signals is important in sleep stage prediction.

**Strengths:**

- S1: The motivation is clear and easy to understand.
- S2: Introducing the superfluous information is interesting.

**Weaknesses:**

- W1: Is the superfluous information minimization effective in the proposed method? In the final loss function in Eq. (10), only the terms corresponding to $I(z_i;d_i)$ and $I(z_i;v_p)$ in Eq. (3) are included.
- W2: Distinguishing the roles of $I(z_i;d_i)$ and $I(z_i;v_i|v_p)$ would make the proposed method more convincing since both terms are explained as domain-relevant information. It would also be beneficial to perform ablation for $I(z_i;d_i)$ and $I(z_i;v_i|v_p)$. Since the superfluous information strongly correlates with $I(z_i;d_i)$, clarifying the efficacy of using not only $I(z_i;d_i)$ but also $I(z_i;v_i|v_p)$ would be intriguing.
- W3: Comparing with the empirical risk minimization (ERM), i.e., merging the datasets of all domains and simply training with a supervised manner without domain labels, would make the result persuasive because Gulrajani et al. [a] reveal that ERM performs the best when the experiment is carefully designed.
- W4: The derivation of the minimal sufficient representation learning would be more understandable if the relationships between ${x}, {v_1}, {v_2}, {z_1}, {z_2}$ are illustrated as a graphical model.
- W5: Displaying the amount of superfluous information and domain-relevant information in Tab. 1 would make the result more convincing.
- W6: Describing the procedure of the proposed method in more detail, e.g., how to compute $H(z_i|d_i)$, would make the paper more easy to follow.
- W7: Although $\lambda_1$ is set to 1 for neglecting $H(z_i)$ in Eq. (25) to derive Eq. (4), checking the effect of $H(z_i)$ would be beneficial.

[a] Ishaan Gulrajani and David Lopez-Paz. "In search of lost domain generalization." ICLR 2021.

**Questions:**

See the weaknesses.

---

> ### Author Response · Authors · 2024-11-23
> **Response to Reviewer 4jps (Part 1)**
>
> We thank the reviewer for these constructive comments, which will greatly help us improve the quality and clarity of our paper.
> > **W1:** Is the superfluous information minimization effective in the proposed method? In the final loss function in Eq. (10), only the terms corresponding to $I(z_i;d_i)$ and  $I(z_i;v_p)$ in Eq. (3) are included.
>
> **Answer of W1:**
>   The $\alpha$ corresponds to the superfluous information minimization effect. As shown in Eqs. (19) and (20), the superfluous information $I(z_i;v_i|v_p)$ can be expressed as $H(z_i |v_p)$.  Consequently, in Eq. (26), the term $\lambda \cdot H(z_i|v_p)$ originates from the superfluous information $I(z_i;v_i|v_p)$,  while the term $H(z_i|v_p)$ itself stems from the mutual information $I(z_i ; v_p)$. In Eq. (11), $\alpha$ can be expressed as $\alpha = \frac{1}{(\lambda+1)\kappa\tau}$  (see Eqs. (31) and (46)). Thus, the effectiveness of reducing superfluous information depends on $\alpha$.
> In our experiments, since $\kappa$ and $\tau$ are treated as constants, we can vary $\alpha$ to observe the effect of minimizing the superfluous information. Specifically, we set $\tau$ = 0.07, and  $\kappa$ is approximated to be around 200 based on [1]. We observed that the model achieves the best performance when $\lambda \approx 70$, which corresponds to $\alpha = 0.001$ in Figure 4.
> We hope this detailed explanation clarifies your concern effectively and provides additional insight into the role of $\alpha$ in our method.
> ***
> > **W2:** Distinguishing the roles of $I(z_i;d_i)$  and $I(z_i;v_i|v_p)$ would make the proposed method more convincing since both terms are explained as domain-relevant information. It would also be beneficial to perform ablation for $I(z_i;d_i)$  and $I(z_i;v_i|v_p)$. Since the superfluous information strongly correlates with $I(z_i;d_i)$, clarifying the efficacy of using not only $I(z_i;d_i)$  but also $I(z_i;v_i|v_p)$ would be intriguing.
>
> **Answer of W2:**
>   To investigate the respective roles of $I(z_i;d_i)$ and $I(z_i;v_i|v_p)$,  we conducted an ablation study on the SleepEDF-20 dataset. The results are presented in the table below. We conducted statistical t-tests between our method and the baseline, yielding a significant p-value ($P<0.001$).
>
> | $I(z_i; d_i)$ | $I(z_i; v_i \mid v_p)$ | **Acc** | **F1** | **$\kappa$** |
> |:-----------------:|:--------------------------:|:---------:|:--------:|:---------:|
> | O               | X                        | 79.1    | 78.1   | 0.712   |
> | X               | O                        | 86.2    | 80.3   | 0.809   |
> | O               | O                        | **86.7**    | **81.1**   | **0.818**   |
>
>   When examining the t-SNE visualization, we observe that when training without  $I(z_i ;  v_i| v_p)$, the features tend to not cluster well within the same class. This is likely due to the influence of $I(z_i;d_i)$, which tends to spread out the features, making it more challenging for the model to learn compact and well-separated representations for each class. These results highlight that if the loss for $I(z_i;d_i)$ is to be used, it is essential to include a minimization of superfluous information term for $I(z_i;v_i |v_p)$, which helps encode meaningful and relevant information within the features. On the other hand, when using only $I(z_i ;  v_i| v_p)$ without $I(z_i;d_i)$, the model’s generalization ability is reduced. We will include this analysis and the corresponding table in the final version to further illustrate the effect of each term on feature representation and model performance.
>
>
>
> ***
> [1] Sra, Suvrit. "A short note on parameter approximation for von Mises-Fisher distributions: and a fast implementation of I s (x)." Computational Statistics 27 (2012): 177-190.

---

> ### Author Response · Authors · 2024-11-23
> **Response to Reviewer 4jps (Part 2)**
>
> > **W3:** Comparing with the empirical risk minimization (ERM), i.e., merging the datasets of all domains and simply training with a supervised manner without domain labels, would make the result persuasive because Gulrajani et al. [a] reveal that ERM performs the best when the experiment is carefully designed.
>
>
> **Answer of W3:**
>     In response to your suggestion, we conducted additional experiments comparing our proposed method with empirical risk minimization (ERM) using the same architecture. The experimental results are as follows, and we have added them to Table 1 in the revised submission.
>
> | Method | SleepEDF-20       |               |               | MASS            |               |               |
> |--------|--------------------|---------------|---------------|-----------------|---------------|---------------|
> |        | **Acc**            | **F1**        | **κ**         | **Acc**         | **F1**        | **κ**         |
> | ERM    | 84.0               | 76.9          | 0.777         | 86.5            | 81.4          | 0.792         |
> | Ours   | **86.7**              | **81.1**          | **0.818**         | **88.3**            | **83.6**          | **0.826**    |
>
> ***
>
> > **W4:** The derivation of the minimal sufficient representation learning would be more understandable if the relationships between $x,v_1,v_2,z_1,z_2$ are illustrated as a graphical model.
>
> **Answer of W4:**
> Thank you for your insightful suggestion. We agree that illustrating the relationships between $x,v_1,v_2,z_1,z_2$ as a graphical model can enhance the understanding of the derivation for minimal sufficient representation learning. In response, we have updated the submission to include the requested graphical model in the Preliminaries section to clarify these relationships and improve the overall clarity of the paper.
> ***
>
> > **W5:** Displaying the amount of superfluous information and domain-relevant information in Tab. 1 would make the result more convincing.
>
> **Answer of W5:**
> Including the amounts of superfluous information and domain-relevant information directly in Table 1 may not be feasible, as it is challenging to estimate these quantities for certain methods. Instead, we will enhance the explanation provided in Figure 6, which illustrates these two information quantities using various methods to offer greater clarity.
>
> We estimate mutual information using the von Mises-Fisher distribution, as described in [2]. This approach is particularly suitable for contrastive learning-based methods, where the latent vector $z$ is normalized. However, since many of the comparative models in Table 1 are not based on contrastive learning, directly adding this information to the table may not be feasible.
> Instead, we will enhance the clarity and detail of Figure 6, which measures the amounts of superfluous information and domain-relevant information. We hope that this enhancement addresses your concerns by providing a more comprehensible and detailed analysis, thereby strengthening the conclusions drawn in our work.
> ***
>
>
>
> > **W6:** Describing the procedure of the proposed method in more detail, e.g., how to compute $H(z_i|d_i)$ would make the paper more easy to follow.
>
> **Answer of W6:** Thank for your insightful comment. We will improve the clarity of the procedure, including how to compute $H(z_i|d_i)$.
>
> We utilized the Stein gradient method[3] to compute $H(z ∣ d)$, following the approach described in previous research [2]. First, we computed the gradient of $H(z ∣ d)$. Using reparameterization, we obtained the score function $\nabla_z \log q(z|d_i) $, which was then approximated using the Stein gradient method. We maximize this gradient to update the model, thereby maximizing  $H(z ∣ d)$. The detailed derivations are provided in Appendix B.2, specifically in Eqs. (31)–(40).
>
>
> ***
> [2] Wen, Liangjian, et al. "MVEB: Self-Supervised Learning With Multi-View Entropy Bottleneck." IEEE Transactions on Pattern Analysis and Machine Intelligence (2024).
> [3] Li, Yingzhen, and Richard E. Turner. "Gradient Estimators for Implicit Models." International Conference on Learning Representations. 2018.

---

> ### Author Response · Authors · 2024-11-23
> **Response to Reviewer 4jps (Part 3)**
>
> > **W7:** Although $\lambda_1$ is set to 1 for neglecting $H(z_i)$ in Eq. (25) to derive Eq. (4), checking the effect of $H(z_i)$ would be beneficial.
>
> **Answer of W7:** We conducted experiments by varying $\lambda_1$ to evaluate its effect on SleepEDF-20. We set the $\alpha = 0.001$, as it was found to be the best value in our previous experiments. The results of these experiments are provided below:
>
> | **λ**   | **Acc** | **F1** | **κ**   |
> |---------|---------|--------|---------|
> | 0.1     | 85.8    | 79.7   | 0.806   |
> | 0.5     | 86.0    | 80.3   | 0.809   |
> | 0.7     | 85.9    | 80.1   | 0.807   |
> | 1       | **86.7**| **81.1**| **0.818**|
> | 1.5     | 86.3    | 80.4   | 0.811   |
> | 2       | 86.2    | 80.2   | 0.810   |
>
> To better explain the results, we provide a mathematical derivation of the objective function without setting $\lambda_1 = 1$.
> When $\lambda_1$ is not fixed to 1, the objective function can be expressed as follows:
>
> $\mathcal{L} (\phi) = \sum_{j}  \sum_{i \in I} \frac{-1}{|P(i)|} \sum_{p \in P(i)} \log \frac{\exp(z_{i,j} \cdot z_{p,j} / \tau)}{\sum_{n \in N(i)} \exp(z_{i,j} \cdot z_{n,j} / \tau)} + \alpha (\lambda_1 - 1) H(z_{i,j}) - \alpha \lambda_1 H(z_{i,j} \mid d_i)$
>
> In the case where $\lambda_1 > 1$, the coefficient in front of $H(z_{i,j})$ is positive, causing the model to attempt to minimize  $H(z_{i,j})$. Minimizing $H(z_{i,j})$ reduces the amount of information contained in $z$, which appears to hinder the learning process.
>
> In the case where $\lambda_1 \le 1$, the model simultaneously maximizes both $H(z_{i,j})$ and  $H(z_{i,j}|d)$. While maximizing $H(z_{i,j}|d)$ minimize the domain-relevant information $I(z_{i,j};d_i)$, maximizing $H(z_{i,j})$ increases $I(z_{i,j};d_i)$, as $I(z;d) = H(z) -H(z|d)$. Therefore, setting $\lambda_1 = 1$ allows the model to focus entirely on maximizing $H(z_{i,j}|d)$, enabling the extraction of more domain-invariant features and improving the model's ability to generalize across domains.

---

> > ### Comment · Reviewer_4jps · 2024-11-26
> >
> > Thank you for your additional response and the new results.
> > While the derivation is indeed well-articulated, I remain concerned about the clarity of the proposed method.
> > Providing a step-by-step explanation for empirically computing $H(z|d)$ and $H(z)$ would be helpful so that readers can easily implement the proposed loss.

---

> > > ### Author Response · Authors · 2024-11-28
> > > **Response to Replying**
> > >
> > > Thank you for your valuable feedback and thoughtful comments. I apologize for any confusion caused by the delay in the revised manuscript upload.
> > >
> > > In the updated version, we have addressed your concerns by including a detailed explanation of the step-by-step process for empirically computing the proposed loss in Appendix B.3 and providing additional clarification below Eq. (10) in the Methods section.
> > >
> > > Additionally, we have incorporated the requested experiments in the Appendix and added a graphical model to the Preliminaries section to improve clarity. Furthermore, we have included a comparison experiment with ERM in Table 1 to enhance the empirical evaluation of our approach.
> > >
> > > Your feedback has been incredibly valuable in improving the manuscript, and we sincerely appreciate your continued guidance and suggestions. We look forward to hearing your thoughts on the revised version.

---

> > > > ### Comment · Reviewer_4jps · 2024-11-28
> > > >
> > > > Thank you for your response and the revised version of the paper.
> > > > My concerns have been addressed.
> > > > I will update my score to 6.

---

### Official Review · Reviewer_iPoE · 2024-10-30

**Soundness:** 2
**Presentation:** 1
**Contribution:** 2
**Rating:** 3
**Confidence:** 5

**Summary:**

The authors propose a new sleep staging method based on contrastive learning called MSMS. This method is inspired by the model SleepPyCo and tries to remove superfluous information from the representation of samples. This should help to reduce the domain shift in the data. The paper proposes several experiments on two classical sleep staging datasets to show the benefits of their methods over competitors.

**Strengths:**

- The paper tries to tackle the distribution shift in sleep staging, which is a crucial issue in the biosignal field.
- The illustrative figures look nice.

**Weaknesses:**

- Figure 1 is a bit hard to understand when not reading the paper
- The paper, in general, lacks clarity.
- Related work is unclear. The authors didn't appropriately introduce the basis of contrastive learning, making understanding their paper hard.
- The notation is a bit messy. For example, in the notation part (which should be before related work), $D_m$ represents all samples for one domain, while after $D$ seems to represent the concatenated domains. After that $D_m$ is never used.
- If adapting between subjects could be challenging, there is a more significant distribution shift between datasets. Several papers deal with adaptation between datasets [1, 2, 3]. Using sleep staging for domain generalization without generalizing it to another dataset does not seem engaging.


[1] Eldele et. al. ADAST: Attentive Cross-domain EEG-based Sleep Staging Framework with Iterative Self-Training, IEEE TETCI, 2022
[2] Gnassounou et. al., Convolutional Monge Mapping Normalization for learning on sleep data, Neurips, 2023
[3] Wang et. al., Generalizable Sleep Staging via Multi-Level Domain Alignment, AAAI, 2024

**Questions:**

- Why did you choose this architecture for your model? Does already existing work inspire you?
- The results of MSMS are slightly above other methods. Did you do a statistical test to see if the improvement is significant?

---

> ### Author Response · Authors · 2024-11-24
> **Response to Reviewer iPoE (Part 1)**
>
> We sincerely appreciate your thoughtful feedback and the effort you put into your review. We hope that our responses below adequately address your concerns.
>
> >**W1:** Figure 1 is a bit hard to understand when not reading the paper
>
> > **W2:** The paper, in general, lacks clarity.
>
> > **W4:** The notation is a bit messy. For example, in the notation part (which should be before related work),  $D_m$ represents all samples for one domain, while after $D$ seems to represent the concatenated domains. After that $D_m$ is never used.
>
> **Answer of W1, W2 and W4:**
> We will improve the clarity of our figures and explanations. Specifically, we will revise Figure 1 to include more descriptive labels, captions, and annotations that succinctly summarize the key ideas conveyed. Additionally, we will revise the notation section to clearly define all symbols and ensure consistency throughout the paper. These changes will address the noted issues and improve the overall coherence of our work.
>
> ***
> > **W3:** Related work is unclear. The authors didn't appropriately introduce the basis of contrastive learning, making understanding their paper hard.
>
> **Answer of W3:**
> Our research focuses on achieving domain generalization in sleep staging, which is reason we structured the related work section around sleep staging and generalization. We acknowledge the importance of providing the foundational basis of contrastive learning. Hence, we included a detailed explanation of contrastive learning and its interpretation from an information-theoretic perspective in the Preliminaries section. To further improve clarity, we will revise this section to provide a more concise and direct explanation of contrastive learning.
>
> ***
>
> >**W5:** If adapting between subjects could be challenging, there is a more significant distribution shift between datasets. Several papers deal with adaptation between datasets [1, 2, 3]. Using sleep staging for domain generalization without generalizing it to another dataset does not seem engaging.
>
> **Answer of W5:**
> Thank you for raising this important point regarding the challenges of domain generalization across datasets. While adapting between datasets is indeed a meaningful challenge, we believe it is substantially important to address the significant variability in bio-signals between subjects. This is because factors such as age, condition, health status, and other individual differences can significantly impact the intensity, patterns of signals, and the distribution of labels [4,5]. Consequently, many existing methods have primarily focused on addressing subject-level variation [6, 7, 8].
> In our experiments, we also applied the method proposed in [3], which addresses dataset-level distribution shifts, to the subject-level variation scenario. As shown in Table 1, its performance was comparatively weaker than that of other domain generalization methods, suggesting that subject-level distribution shifts pose distinct and significant challenges.
>
> ***
> [1] Eldele et. al. ADAST: Attentive Cross-domain EEG-based Sleep Staging Framework with Iterative Self-Training, IEEE TETCI, 2022
> [2] Gnassounou et. al., Convolutional Monge Mapping Normalization for learning on sleep data, Neurips, 2023
> [3] Wang et. al., Generalizable Sleep Staging via Multi-Level Domain Alignment, AAAI, 2024
> [4] Buysse, Daniel J., et al. "EEG spectral analysis in primary insomnia: NREM period effects and sex differences." *Sleep*, 2008
> [5] Ohayon, Maurice M., et al. Meta-analysis of quantitative sleep parameters from childhood to old age in healthy individuals: developing normative sleep values across the human lifespan, *Sleep*, 2004
> [6] Lee, Seungyeon, Thai-Hoang Pham, and Ping Zhang. DREAM: Domain Invariant and Contrastive Representation for Sleep Dynamics, ICDM, 2022.
> [7] Yang, Chaoqi, M. Brandon Westover, and Jimeng Sun. ManyDG: Many-domain Generalization for Healthcare Applications, ICLR 2023.
> [8] Wang, Yihe, et al. Contrast everything: A hierarchical contrastive framework for medical time-series, Neurips, 2024

---

> ### Author Response · Authors · 2024-11-24
> **Response to Reviewer iPoE (Part 2)**
>
> >**Q1:** Why did you choose this architecture for your model? Does already existing work inspire you?
>
> **Answer of Q1:**
> Thank you for your insightful question. We will provide a clearer explanation of our model architecture selection in the revised version.
> Our model architecture was inspired by the work proposed in [9], which was specifically designed to support contrastive learning and leverage multi-scale features. This design aligns well with the objectives of our method, as it complements contrastive learning while effectively integrating multi-scale representation learning to capture diverse temporal and spectral characteristics.
>
> ***
> >**Q2:** The results of MSMS are slightly above other methods. Did you do a statistical test to see if the improvement is significant?
>
>
> **Answer of Q2:**
> Yes, we conducted statistical t-testing between our method and the baseline, calculating a p-value (P 0< .001) on both SleepEDF-20 and MASS datasets. We employed a k-fold approach for evaluation, ensuring that each subject was included as the test set exactly once. Specifically, on the SleepEDF-20 dataset, our method achieved a variation of 2.06, whereas the second-best model, SleePyCo [9], exhibited a higher variation of 3.13. On the MASS dataset, our method's variation was 3.03, compared to SleePyCo's 5.23. These results indicate that our method not only slightly improves average performance but also provides more stable and reliable outcomes across different folds.
>
> ***
> [9] Lee, Seongju, et al. "Sleepyco: Automatic sleep scoring with feature pyramid and contrastive learning." Expert Systems with Applications 240 (2024): 122551.

---

> ### Comment · Reviewer_iPoE · 2024-11-26
>
> I thank the authors for their reply to my review. I did not understand if the updates were already available in the pdf file. It seems not.
>
> If the topics dealt with in the paper are crucial in the community, as said in the responses (tackle shift coming from age, condition ...), the paper still lacks clarity and critical experiments (i.e., cross dataset experiment). At this point, the paper does not bring much to the existing literature.
>
> I will keep my score, but I encourage the authors to continue researching this exciting topic, which is not solved right now.

---

> > ### Author Response · Authors · 2024-11-28
> > **Response to Replying**
> >
> > Thank you for your insightful feedback and constructive comments. I apologize for any confusion that may have arisen due to the delay in submitting the revised manuscript.
> >
> > To address the points you raised, we have revised Figure 1 and its caption to enhance clarity and updated the notations throughout the manuscript. Additionally, we have included an explanation of contrastive learning to improve the conceptual understanding of the proposed approach.
> >
> > As you highlighted the importance of addressing domain shift issues across datasets, we have acknowledged this limitation and included a discussion in the Conclusion as a direction for future work.
> >
> > Your feedback has been invaluable in refining our research, and we sincerely appreciate your constructive input.

---

> > > ### Comment · Reviewer_iPoE · 2024-12-02
> > >
> > > Thank you for the new version of the pdf. If the paper is going in the right direction, I think the paper still lacks clarity and novelty.
> > >
> > > I encourage you to keep up your efforts to improve the clarity of your paper and implement the cross-dataset experiment that could enhance your work and give meaningful insight for the next.

---

### Official Review · Reviewer_kBq5 · 2024-10-31

**Soundness:** 2
**Presentation:** 3
**Contribution:** 2
**Rating:** 3
**Confidence:** 5

**Summary:**

This paper solves the domain adaptation problem in the context of sleep stage classification using EEG. In particular, the authors proposed  a Multi-Scale Minimal Sufficient representation learning (MSMS) framework that reduces domain-relevant information while preserving essential temporal and spectral features. Evaluations are conducted on publicly available sleep staging benchmark datasets, SleepEDF-20 and MASS to demonstrate the effectiveness of the proposed method.

**Strengths:**

* This problem the authors aimed to solve is practical.
* This paper is easy to follow.

**Weaknesses:**

* a) Limited novelty. Although the authors spent majority of the space talking about mutual information (MI) and superfluous information. They ended up adding an extra term of domain conditioned entropy in addition to the well known contrastive loss. Adding regularisation in the context of domain adaptation is not new. For instance, Domain-Adversarial Neural Networks (DANN) added an inverse gradient to confuse features extracted from two domains. The so called MULTI-SCALE is essentially applying contrastive loss to intermediate features which is not new in both contrastive learning or domain adaptation.

* b) Poor clarity. The authors spent too much space on MI which is in the end intractable. Instead, the authors should put more focus on describing the details of how exactly their framework works. For instance, it seems that the proposed framework is a dual-stage framework, how is the second stage trained and using what data ? How is H(z|d) calculated? What is the network structure used ?

* c) Marginal improvement. As seen in Table 1, on SleepEDF-20, the improvement over the second best approach is 0.4%. On MASS, the improvement is 0.3%.

* d) Insufficient ablation studies. The authors proposed two novelties in this work, a regularisation term and the application of the loss function (which layers). These should be studied individually to see the contribution of each.

**Questions:**

See weakness.

---

> ### Author Response · Authors · 2024-11-24
> **Response to Reviewer kBq5 (Part 1)**
>
> Thank you for dedicating your time and effort to providing such constructive feedback.
>
> > **a):**  Limited novelty. Although the authors spent majority of the space talking about mutual information (MI) and superfluous information. They ended up adding an extra term of domain conditioned entropy in addition to the well known contrastive loss. Adding regularisation in the context of domain adaptation is not new. For instance, Domain-Adversarial Neural Networks (DANN) added an inverse gradient to confuse features extracted from two domains. The so called MULTI-SCALE is essentially applying contrastive loss to intermediate features which is not new in both contrastive learning or domain adaptation.
>
> **Answer of a):**
> We sincerely appreciate your thoughtful feedback. We will include additional explanations regarding the novelty of our work.
>
> + **Novelty of minimal sufficient learning for domain generalization:**
> While regularization techniques have been used in domain generalization, our approach introduces a regularization term that is theoretically grounded and can be seamlessly integrated into contrastive learning without the need for additional modules or layers, which we believe constitutes a novel contribution. In contrast, traditional approaches such as Domain-Adversarial Neural Networks (DANN) rely on additional domain classifiers, making them dependent on adversarial training. This distinction highlights the unique advantage of our method in achieving domain invariance in a simpler and more efficient manner.
>
> + **Novelty of multi-scale learning:**
> Our multi-scale learning approach is novel in its ability to complement minimal sufficient representation learning by preventing information loss during feature extraction. Minimal sufficient representation learning inherently reduces the amount of information, and when applied solely to specific layers, it can cause the model to overly focus on features extracted from those layers. As demonstrated in the ablation study results (Figure 3 and Table 2), applying minimal sufficient learning without incorporating multi-scale learning (focusing solely on high-level information) leads to performance degradation. In contrast, combining multi-scale learning with contrastive learning results in decreased performance. These findings highlight that multi-scale learning must be thoughtfully integrated with minimal sufficient representation learning.
>
> ***
> > **b):** Poor clarity. The authors spent too much space on MI which is in the end intractable. Instead, the authors should put more focus on describing the details of how exactly their framework works. For instance, it seems that the proposed framework is a dual-stage framework, how is the second stage trained and using what data? How is $H(z|d)$ calculated? What is the network structure used?
>
> **Answer of b):**
> Thank you for your valuable comments. We will update further details on the sleep staging process and how to calculate the conditional entropy $H(z|d)$.
>
> + **Sleep Staging Process:**
> We recognize that the explanation regarding the sleep staging method design may have been insufficient, potentially causing some confusion. In our work, we follow a two-stage framework, and architectural design is based on prior work [1].  In the second stage, a sequence of $L$ EEG signals is fed into a transformer-based architecture, which computes a weighted sum to produce the predicted label $\hat{y}$, where $L$ is the sequence length. The multi-scale cross entropy loss is used as the objective function, as detailed in the sleep staging section of the Appendix B.5. The data used for this stage is the same as that used during the pre-training phase.
>
>
> + **How is $H(z|d)$ calculated?**
> We utilized the Stein gradient method to compute $H(z∣d)$, following the approach described in previous research [2]. First, we computed the gradient of $H(z∣d)$. Using reparameterization, we obtained the score function $\nabla_z \log q(z|d) $, which was then approximated using the Stein gradient method [3]. We maximize this gradient to update the model, thereby maximizing  $H(z ∣ d)$. The detailed derivations are provided in Appendix B.2, specifically in Eqs. (31)–(40).
>
> ***
> [1] Lee, Seongju, et al. "Sleepyco: Automatic sleep scoring with feature pyramid and contrastive learning." *Expert Systems with Applications* 240 (2024): 122551.
>
> [2] Wen, Liangjian, et al. "MVEB: Self-Supervised Learning With Multi-View Entropy Bottleneck." *IEEE Transactions on Pattern Analysis and Machine Intelligence* (2024).
>
> [3] Li, Yingzhen, and Richard E. Turner. "Gradient Estimators for Implicit Models." International Conference on Learning Representations. 2018.

---

> ### Author Response · Authors · 2024-11-24
> **Response to Reviewer kBq5 (Part 2)**
>
> >**c):** Marginal improvement. As seen in Table 1, on SleepEDF-20, the improvement over the second best approach is 0.4%. On MASS, the improvement is 0.3%.
>
> **Answer of c):**
> While the performance improvement may appear marginal, our method consistently exhibits lower variation across all out-of-distribution (OOD) settings, highlighting its robustness in handling diverse and unseen domains. We employed a k-fold approach for evaluation, ensuring that each subject was included as the test set exactly once. Specifically, on the SleepEDF-20 dataset, our method achieved a variation of 2.06, whereas the second-best model, SleePyCo [1], exhibited a higher variation of 3.13. On the MASS dataset, our method's variation was 3.03, compared to SleePyCo's 5.23. Furthermore, we conducted statistical t-testing between our method and the baseline, calculating a p-value (P 0< .001) on both SleepEDF-20 and MASS datasets.
> These results indicate that our method not only slightly improves average performance but also provides more stable and reliable outcomes across different folds.
>
> ***
> >**d):** Insufficient ablation studies. The authors proposed two novelties in this work, a regularisation term and the application of the loss function (which layers). These should be studied individually to see the contribution of each.
>
> **Answer of d):**
> We conducted ablation studies, the results of which are detailed in Section 5.4 (Ablation Study) and illustrated in Figure 3. To enhance clarity, we will update the chart by replacing the label "SCL" with "Ours (w/o regularization)" to better reflect the context.
> In this ablation study, we observed that the best performance was achieved only when both methods were applied together, while neither method independently results in significant performance improvement. This can be attributed to the nature of minimal sufficient learning, which reduces the information content in features. Without the application of multi-scale learning, the model risks overfitting to information from specific layers. As shown in Table 3, when minimal sufficient learning is applied exclusively to high-level features, the model tends to focus predominantly on wake stages. In the context of sleep staging, it is well-established that high-level features are particularly effective for distinguishing wake stage characteristics  [1]. These findings underscore the importance of combining minimal sufficient learning with multi-scale learning to achieve balanced and generalized performance.
>
>
>
> ***
> [1] Lee, Seongju, et al. "Sleepyco: Automatic sleep scoring with feature pyramid and contrastive learning." Expert Systems with Applications 240 (2024): 122551.

---

> > ### Comment · Reviewer_kBq5 · 2024-11-27
> >
> > Thank you for providing a response to my concerns. While I acknowledge the importance and value of the problem your work seeks to address, I must note that significant issues with clarity and novelty remain, as also highlighted by other reviewers. The methodology and presentation, in particular, would benefit from substantial refinement to better communicate the contributions and improve the overall impact of the work. As such, I will maintain my current rating at this time.

---

> > > ### Author Response · Authors · 2024-11-28
> > > **Response to Replying**
> > >
> > > Thank you for your valuable feedback and thoughtful comments. I apologize for any confusion caused by the delay in the revised manuscript upload.
> > >
> > >
> > >
> > > As per your suggestions, we have revised the manuscript to improve clarity. Specifically, we have provided a detailed explanation of how $H(z∣d)$ is calculated and included additional descriptions and visuals to better illustrate the overall framework (Figure 3). Furthermore, we have refined the presentation of the ablation study results, revising both the text and the accompanying figures to enhance clarity and precision.
> > >
> > > Your constructive feedback has been invaluable in improving the quality of our research, and we sincerely appreciate your thoughtful input.

---

### Official Review · Reviewer_93jn · 2024-11-03

**Soundness:** 3
**Presentation:** 2
**Contribution:** 2
**Rating:** 5
**Confidence:** 5

**Summary:**

This paper proposed domain generalization method for sleep staging analysis automatically. The main way to learn domain-invariant features are through contrastive learning however the problem of existing methods is unable to reduce domain relevant information embedded in the features. In addition, the claimed challenge is existing methods neglect multi-scale nature of sleep signals, missing temporal and spectral features. The paper proposes a multi-scale minimal sufficient representation learning framework. This framework has two functions, including the domain-relevant information reduction; and the temporal and spectral features for sleep stage classification.

**Strengths:**

-	The method and theorem seem correct
-	The writing and organization seem clear

**Weaknesses:**

-	One of the two main issues is the unclear motivations. In the introduction, especially the fig1a and 1b, it is ambiguous and unclear to understand your proposals. How to estimate the real and genuinely domain-invariant representations? This needs very clear and exact computations or experimental proofs, rather than using the toy-example figures.

-	The second of the two main issues is in the novelty. The information bottleneck combined with the contrastive learning has been widely and commonly used in general machine learning and representation learning. It is clear that this solution may combined the existing work and method into the sleep staging. In addition, how to handle the genuine sleep staging characteristics in the method design is missing. Integration of existing method into the application is the major issue.

**Questions:**

see weaknesses and answer please item by item.

---

> ### Author Response · Authors · 2024-11-23
> **Response to Reviewer 93jn (Part 1)**
>
> Thank you for your valuable and insightful comments. We sincerely appreciate your time and effort in providing constructive feedback.
>
> > **W1:** One of the two main issues is the unclear motivations. In the introduction, especially the fig1a and 1b, it is ambiguous and unclear to understand your proposals. How to estimate the real and genuinely domain-invariant representations? This needs very clear and exact computations or experimental proofs, rather than using the toy-example figures.
>
> **Answer of W1**:
>   We will improve the clarity of our novelty and the presentation of Figures 1a and 1b to ensure they better communicate our proposals. Additionally, we will clearly specify how domain-invariant representations are estimated.
>
>  - **Unclear motivations**:
>   The primary motivation for our work is to minimize domain-relevant information by reducing superfluous information, addressing the limitation of existing contrastive learning methods that inherently allow domain-relevant characteristics to remain in the features.
>    Traditional contrastive learning approaches often leave domain-relevant features embedded within the superfluous information, as illustrated in Figure 1(a) under the label "excess domain-relevant information." This embedding of domain-specific information into the features hinders their ability to generalize effectively to new domains, resulting in degraded performance. To address this limitation, our proposed method explicitly reduces superfluous information to remove excess domain-relevant information. As shown in Figure 1(b), this reduction is represented by the decrease in the overlap between the feature $z_i$ and the domain information $d_i$, indicating that the learned features have become more domain-invariant. This observation is further supported by Theorem 1, which demonstrates that reducing superfluous information leads to features that are more domain-invariant.
>
>
>  - **How to estimate the real and genuinely domain-invariant representations?:**
>   We estimate the domain-relevant information $I(z;d)$ using the von Mises-Fisher distribution and Stein gradient estimation [1], as proposed in [2], where a lower $I(z;d)$ value indicates a more domain-invariant feature. The term $I(z;d)$ can be decompose as $H(z) - H(z|d)$, where the entropy $H(z)$ is expressed as $\mathbb{E}[-\log p(z)]$. We can estimate $I(z;d)$ by accurately approximating estimating $p(z)$. We use the von Mises-Fisher distribution, which models distributions on a hypersphere, to approximate $p(z)$, as $z$ is normalized and lies on a hypersphere.
>   The results of estimating $I(z;d)$  using this method are presented in Figure 7. The experimental findings demonstrate that the proposed approach achieves lower  $I(z;d)$  values compared to baseline methods. This indicates that our method effectively extracts domain-invariant features, further validating its robustness in addressing domain generalization challenges.
>
>
>
> ***
> [1] Li, Yingzhen, and Richard E. Turner. "Gradient Estimators for Implicit Models." *International Conference on Learning Representations* (2018).
> [2] Wen, Liangjian, et al. "MVEB: Self-Supervised Learning With Multi-View Entropy Bottleneck." *IEEE Transactions on Pattern Analysis and Machine Intelligence* (2024).

---

> ### Author Response · Authors · 2024-11-23
> **Response to Reviewer 93jn (Part 2)**
>
> >**W2:** The second of the two main issues is in the novelty. The information bottleneck combined with the contrastive learning has been widely and commonly used in general machine learning and representation learning. It is clear that this solution may combined the existing work and method into the sleep staging. In addition, how to handle the genuine sleep staging characteristics in the method design is missing. Integration of existing method into the application is the major issue.
>
>
>
> **Answer of W2:**
>   + **Novelty of minimal sufficient learning for domain generalization:**
>      We believe our work demonstrates clear novelty as it identifies the differences from existing methods and addresses the limitations of prior research by proposing a solid theoretical foundation to overcome these challenges.
> While we acknowledge that our method incorporates aspects of prevalent contrastive learning, our approach introduces a key difference by explicitly addressing domain-relevant information  $I(\boldsymbol{z};d)$. Specifically, we extend the general objective described in Eq. (1), based on bottleneck information theory, by explicitly minimizing this domain-relevant information, as shown in Eq. (2). Consequently, we derived a novel regularization term that maximizes the conditional entropy $H(z|d)$, ensuring the domain generalized performance.
>  To the best of our knowledge, we are the first to address the domain information contained in the superfluous information and to explicitly address the domain information embedded within superfluous information, providing a unique contribution to the field and advancing the theoretical and practical understanding of domain generalization.
>
>   + **How to handle the genuine sleep staging characteristics in the method design:**
>       Our method employs minimal sufficient representation learning on features extracted from multiple layers of the encoder to effectively capture the unique characteristics of sleep stages. Each layer targets complementary temporal and spectral information, as outlined in [3]. Specifically, lower layers emphasize long-term temporal dependencies, crucial for identifying slow-wave sleep (N3) characterized by low-frequency activity (e.g., 0.5–2 Hz). In contrast, higher layers focus on high-frequency patterns, such as beta (13–30 Hz) and alpha (8–13 Hz) rhythms, which are dominant in a Wake (W) stage. These multi-layer features are then passed into a transformer-based classifier to predict labels, as detailed in Appendix B.5.
>      Multi-scale learning, while addressing the limitations of minimal sufficient learning and contributing to the novelty of our research, is not independently effective in the context of sleep data. As demonstrated in the ablation study results (Figure 4 and Table 2), combining multi-scale learning with contrastive learning results in decreased performance. Similarly, applying minimal sufficient learning without incorporating multi-scale learning (focusing solely on high-level information) leads to performance degradation. This degradation occurs because the model becomes overly reliant on information from specific layers. These findings underscore the necessity of combining multi-scale and minimal sufficient representation learning to effectively capture diverse temporal and spectral characteristics while maintaining robust domain generalization.
>
> ***
> [3] Lee, Seongju, et al. "Sleepyco: Automatic sleep scoring with feature pyramid and contrastive learning." Expert Systems with Applications 240 (2024): 122551.

---

### Comment · Area_Chair_kXrE · 2024-11-26
**Encouragement to Actively Participate in the Discussion Phase**

Dear Reviewers,

Thank you for your valuable contributions to the review process so far. As we enter the discussion phase, I encourage you to actively engage with the authors and your fellow reviewers. This is a critical opportunity to clarify any open questions, address potential misunderstandings, and ensure that all perspectives are thoroughly considered.

Your thoughtful input during this stage is greatly appreciated and is essential for maintaining the rigor and fairness of the review process.

Thank you for your efforts and dedication.

---

### Author Response · Authors · 2024-11-28
**Updating Manuscript**

We sincerely appreciate the reviewers' thoughtful comments and feedback.

To address suggestions, we have thoroughly revised the manuscript, performed additional experiments, and incorporated new tables.
We have made significant efforts to enhance the clarity and highlight the novelty of the work.

We have uploaded the latest revised version, with **blue** markings indicating new or modified content.

Here is a quick summary of the revisions:

1. Edited introduction to explain the motivation and novelty.
2. Edited Figure 1 and its caption to enhance the clarity of the motivation and novelty.
3. Improved the clarity of notation throughout the manuscript.
4. A detailed explanation of how $H(z∣d)$ is calculated is included in Appendix B.3 and the methods section.
5. Figure 3 and the caption have been revised to better represent the overall framework.
6. Created a new "Overall Framework" section under the Methods to describe the general approach to sleep staging.
7. Added an explanation of contrastive learning and a graphical model to the preliminaries section.
8. Added comparison experiments with ERM to Table 1.
9. Revised expressions in the ablation study section for improved clarity.
10. A discussion on addressing dataset distribution shifts as a future direction was added in the Conclusion.
11. Additional experiments requested by reviewer 4jps are included in the Appendix.

---

### Note · Authors · 2024-12-02

**Comment:**

Thank you for your valuable comments and the interest you have shown in our work.
We deeply appreciate the feedback provided, which has been highly insightful.


After careful consideration, we have decided to withdraw this manuscript to further refine and improve the content based on the feedback received.

Thank you once again for your time and thoughtful review.

**Withdrawal Confirmation:**

I have read and agree with the venue's withdrawal policy on behalf of myself and my co-authors.